# Spin mapping of intralayer antiferromagnetism and field-induced spin reorientation in monolayer CrTe₂

Jing-Jing Xian[1,6], Cong Wang [2,6], Jin-Hua Nie[1,6], Rui Li[1], Mengjiao Han[3,4], Junhao Lin [3,4], Wen-Hao Zhang [1], Zhen-Yu Liu[1], Zhi-Mo Zhang[1], Mao-Peng Miao[1], Yangfan Yi[5], Shiwei Wu [5], Xiaodie Chen[1], Junbo Han[1], Zhengcai Xia[1], Wei Ji [2✉] & Ying-Shuang Fu [1✉]

Intrinsic antiferromagnetism in van der Waals (vdW) monolayer (ML) crystals enriches our understanding of two-dimensional (2D) magnetic orders and presents several advantages over ferromagnetism in spintronic applications. However, studies of 2D intrinsic antiferromagnetism are sparse, owing to the lack of net magnetisation. Here, by combining spin-polarised scanning tunnelling microscopy and first-principles calculations, we investigate the magnetism of vdW ML CrTe₂, which has been successfully grown through molecular-beam epitaxy. We observe a stable antiferromagnetic (AFM) order at the atomic scale in the ML crystal, whose bulk is ferromagnetic, and correlate its imaged zigzag spin texture with the atomic lattice structure. The AFM order exhibits an intriguing noncollinear spin reorientation under magnetic fields, consistent with its calculated moderate magnetic anisotropy. The findings of this study demonstrate the intricacy of 2D vdW magnetic materials and pave the way for their in-depth analysis.

[1] School of Physics and Wuhan National High Magnetic Field Center, Huazhong University of Science and Technology, Wuhan 430074, China. [2] Beijing Key Laboratory of Optoelectronic Functional Materials and Micro-Nano Devices, Department of Physics, Renmin University of China, Beijing 100872, China. [3] Department of Physics, Southern University of Science and Technology, Shenzhen 518055, China. [4] Shenzhen Key Laboratory for Advanced Quantum Functional Materials and Devices, Southern University of Science and Technology, Shenzhen 518055, China. [5] State Key Laboratory of Surface Physics, Department of Physics, Fudan University, Shanghai 200433, China. [6] These authors contributed equally: Jing-Jing Xian, Cong Wang, Jin-Hua Nie. ✉email: wji@ruc.edu.cn; yfu@hust.edu.cn

The exploration of the long-range magnetic order down to the single-layer limit is not only of fundamental importance for examining its existence at finite temperatures but also driven by the technological requirements of miniaturised device circuits. According to the Mermin–Wagner theorem[1], the long-range magnetic order cannot exist in one and two dimensions at finite temperatures owing to the low energy cost of fluctuations, which benefit the total energy of the system via entropy increase. However, in the presence of magnetic anisotropy, the fluctuation effect can be effectively suppressed, and the long-range magnetic order revives. The recent discovery of intrinsic ferromagnetism at ultrathin van der Waals (vdW) films of $CrI_3$[2] and $Cr_2Ge_2Te_6$[3] have spurred considerable interest in seeking magnetism in single layers and their manipulations[4–9].

Antiferromagnetic (AFM) orders have gained increasing attention in spintronic applications owing to their advantages over their ferromagnetic counterpart in achieving ultrafast dynamics, large magnetoresistance transport, and immunity to magnetic field disturbance associated with absent stray fields[10–14]. However, despite the fruitful research on two-dimensional (2D) ferromagnetism, AFM orders at the 2D limit are significantly less explored because of the vanishing net magnetisation, causing difficulty in detection. Pioneering studies on Raman spectroscopy have identified AFM orders in a vdW monolayer (ML) $FePS_3$ via spin-phonon interactions[15,16]. As AFM orders possess various possible configurations, their specific spin textures are urged to be determined with real-space spin-sensitive probes. Spin-polarised scanning tunnelling microscopy (SPSTM), performed by recording spin-dependent electron tunnelling currents, can directly identify the magnetic order with atomic-scale spin resolution[17], enabling considerable progress in the studies on magnetism in vdW materials[18–21]. Particularly, a double-stripe AFM spin structure, coexisting with superconductivity, was observed on a $Fe_{1+y}Te/Bi_2Te_3$ heterojunction via SPSTM[21]. However, such superconductivity is believed to be incurred from prominent interfacial interactions, thus casting doubt on whether the AFM order in $Fe_{1+y}Te$ is intrinsically stable.

For bulk $CrTe_2$ crystals, a ferromagnetic ground state with a Curie temperature of 310 K was previously identified[22], rendering it a promising candidate for exploring high-Curie-temperature ML ferromagnets. A recent study found that the Curie temperature of a ~10 ML $CrTe_2$ exhibits ferromagnetism at room temperature[23]. However, theoretical studies predicted that the magnetism of vdW MLs varies under a few external perturbations, including strain[24–26] and doping or interlayer coupling[27,28], leaving their magnetic orders at the ML limit as an open issue.

Here, we report the successful growth of $CrTe_2$ films on a SiC-supported bilayer graphene substrate and achieve spin-resolved imaging of the ML $CrTe_2$ with SPSTM. Although $CrTe_2$ has strong itinerant ferromagnetism in bulk[22], its ML surprisingly reveals an AFM spin texture at the atomic scale. The AFM order is of zigzag type with the magnetic easy axis in the $y$-$z$ plane and 70° off the $z$ axis. With a magnetic field applied perpendicular to the basal plane, the ML $CrTe_2$ exhibits a noncollinear spin reorientation transition, which, to our knowledge, is the first report of such a real-space observation. Our work paves the way for in-depth studies on the fundamental physics of 2D antiferromagnetism and sets a foundation for its application in AFM spintronics.

## Results and discussion

Because of the multivalent nature of the Cr cation, chromium tellurides have multiple polymorphs[29], posing a challenge to grow single-phase compounds. With the judicious tuning of the flux ratio of Cr and Te and the substrate temperature, we successfully grew uniform $CrTe_2$ films. $CrTe_2$ has a trigonal layered structure belonging to space group P3m1[22]. Each vdW layer of $CrTe_2$ constitutes chromium cations centring at tellurium octahedrons, forming a 1 T structure (Fig. 1a). Figure 1b shows a typical scanning tunnelling microscopy (STM) topographic image of the as-grown thin films, which are dominated with an ML and a second layer, with a fraction of a third layer. The second-layer film manifests an apparent height of $0.63 \pm 0.16$ nm, which is insensitive to the bias and is consistent with the theoretical ML height (0.6168 nm). However, the apparent height of the first-layer film varies between 1.05 nm and 0.88 nm at different biases (Supplementary Fig. S1), reflecting a large vdW gap at the interface and the distinct electronic structure between the graphene and $CrTe_2$ (Supplementary Fig. S2).

To disentangle the electronic effect on the structural determination, the $CrTe_2$ film is characterised with cross-sectional scanning transmission electron microscopy (STEM). A high-angle annular dark-field (HAADF) STEM image (Fig. 1c) confirms the 1 T structure of the bilayer $CrTe_2$ with Te capping. Owing to its heavier mass, Te has brighter contrast than Cr does. The associated integrated differential phase contrast (iDPC) image (Fig. 1d) resolves the fine details of the film, clearly showing the $CrTe_2$ layer and the supporting graphene/SiC substrate, where the graphene layer is distinguished by its weaker contrast and larger interlayer spacing than the underneath SiC lattice. The heights of the second- and first-layer $CrTe_2$ are estimated as 0.6 nm and 0.8 nm, respectively, conforming to the STM measurement. The negligible contrast between two $CrTe_2$ layers in Fig. 1d demonstrates clearly a vdW gap without intercalated Cr atoms. The vdW gap between graphene and $CrTe_2$ is also clearly seen, which is larger than the interlayer vdW gap in $CrTe_2$. The elemental composition of the film is 1:2 for the Cr:Te ratio according to the standard-based quantification analysis of the electron energy loss spectrum (EELS) line scan across the interface (Fig. 1e). Moreover, the energy-dispersive spectroscopy (EDS) mapping (Supplementary Fig. S3) further verifies the Cr and Te elemental distributions in each layer.

An atomic-resolution STM image of the ML film (Fig. 1f) and its associated fast Fourier transform (FFT) image (Fig. 1g) reveal an isosceles lattice with in-plane constants measured as $a_1 = 3.4$ Å, $a_2 = 3.7$ Å, and $a_3 = 3.7$ Å, which obviously deviates from an expected regular one of the bulk surfaces. The isosceles lattice is further confirmed by the STM imaging of a domain boundary, which expresses a mirror symmetry between the two neighbouring domains (Supplementary Fig. S4). An evident $2 \times 1$ periodicity is exhibited in the atomic-resolution image and the FFT image of the ML $CrTe_2$, which is closely related to its AFM spin texture, as elucidated later. Six additional Bragg peaks are present around the central zone (Fig. 1g, yellow rectangle), which stem from a 6×6 Moiré pattern at the graphene/SiC(0001) interface[30]. On the second-layer film, the $2 \times 1$ structure exhibits mixing domains of three equivalent orientations (Fig. 1h).

SPSTM measurement was applied to the $CrTe_2$ ML with a Cr-coated W tip, where the tip magnetisation is canted. Because of its small stray field, the AFM Cr tip barely perturbs the magnetisation of the sample and is resistant against the magnetic field. An SPSTM image at $-0.1$ V (Fig. 2a) shows a clear zigzag-like pattern that is distinct from the $2 \times 1$ structure shown in Fig. 1f. Its magnetic unit cell (Fig. 2a, white rectangle) is twice the original lattice in both directions. As is seen from its FFT image (Fig. 2b), the diffraction spots (white circles) of the zigzag pattern, in comparison with those in Fig. 1g, are coexisting with those of the $2 \times 1$ structure (yellow circles) and are orthogonal to one another. The orthogonality is substantiated from the image of the same area at $-60$ mV (Supplementary Fig. S5), where the $2 \times 1$ structure

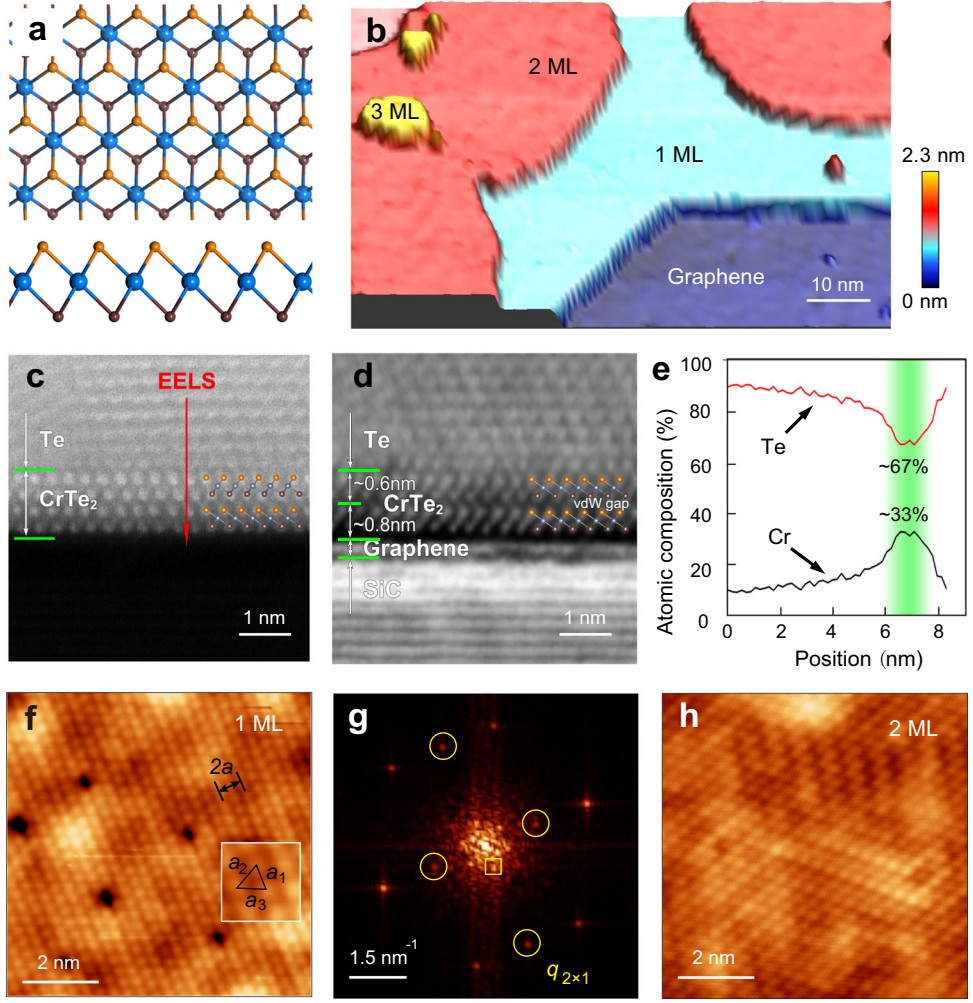

**Fig. 1 Morphology of 1T-CrTe$_2$ films. a** Top view (up) and side view (down) of the crystal structure of 1T-CrTe$_2$. The orange (brown) balls represent the top (bottom) Te atoms, and the dodger-blue balls represent the Cr atoms. **b** Pseudo-3D topographic STM image ($V_b = 2$ V, $I_t = 10$ pA) of CrTe$_2$ films. The thicknesses of the films are marked. **c**, **d** Atomic resolution of the HAADF-STEM (**c**) and iDPC images (**d**) of the side view of 2 ML CrTe$_2$. **e** EELS quantification analysis along the red line in **c**, demonstrating that the chemical stoichiometry of Cr and Te is 1:2 at the green-shaded region. **f**, **g** Atomic resolution of the STM image ($V_b = 0.1$ V, $I_t = 85$ pA) of 1 ML CrTe$_2$ (**f**) and its FFT image (**g**). Inset of (**f**) shows a zoomed-in view ($1 \times 1$ nm$^2$) of the ML CrTe$_2$. The periodicity of the $2 \times 1$ structure is marked. The yellow circles in (**g**) mark the $2 \times 1$ superstructure of the Te lattice. **h** Atomic resolution of the STM image ($V_b = -25$ mV, $I_t = 10$ pA) of 2 ML CrTe$_2$.

becomes more visible. Because the zigzag pattern is only resolvable with the magnetic tip, it is ascribed as a spin contrast.

Such ascription is further confirmed by applying magnetic fields perpendicular to the basal plane of the films, i.e., the $z$ direction. Figure 2c, d depict two SPSTM images of the same area taken at 1 T and −1 T, respectively. There is a defect acting as a marker (green circle). Intriguingly, the zigzag pattern concertedly reverses its contrast relative to the defect under the opposite field, thereby unambiguously proving its origin from the spin contrast. The spin contrast becomes more enhanced in the differential image (Fig. 2e) between Fig. 2c, d and vanishes in their sum image (Fig. 2f), resembling a spin-averaged image shown in Fig. 1f. Such a spin-contrast reversal can also be clearly seen from their line profiles (Fig. 2g) extracted along with the same locations of Fig. 2c–f. The spin contrast keeps its registration with the defect marker in both field orientations up to 5 T (Supplementary Fig. S6) and memorises the history of the field orientation upon its removal (Supplementary Fig. S7). The threshold field is small (<0.2 T) to induce the spin-contrast reversal and is also highly reproducible measured with different Cr tips (Supplementary Fig. S8). For Cr tips, their magnetisation conventionally persists

more than 2 T and varies for different tips[17]. In this regard, the spin-contrast reversal is unlikely arisen from the tip but from the CrTe$_2$ ML, and the magnetisation direction of the tip sustains up to 5 T. The spin contrast directly evidences the existence of an AFM magnetic order. Moreover, the zigzag spin contrast of different domains of ML CrTe$_2$ is not identical (Supplementary Fig. S9), suggesting that the tip magnetisation is canted with finite in-plane spin sensitivity (Supplementary Note), as is also substantiated from the spin mapping with a Fe-coated tip[17] (Supplementary Fig. S10). It is noted that no spin contrast is observed in the second layer CrTe$_2$ at present, whose origin should be investigated in future.

We perform density functional theory (DFT) calculations to unveil the origin of the spin contrast and its field-induced switching. A zigzag AFM state is the most favoured among all considered magnetic configurations (Supplementary Fig. S11) in the fully relaxed freestanding CrTe$_2$ ML (Supplementary Table S1). The geometry of this phase breaks the three-fold rotational symmetry and shows a 0.1 Å shrink for the lattice constant along with directions $a_1$ and $a_2$, i.e., from 3.70 Å to 3.60 Å (Supplementary Table S1), which is slightly smaller than

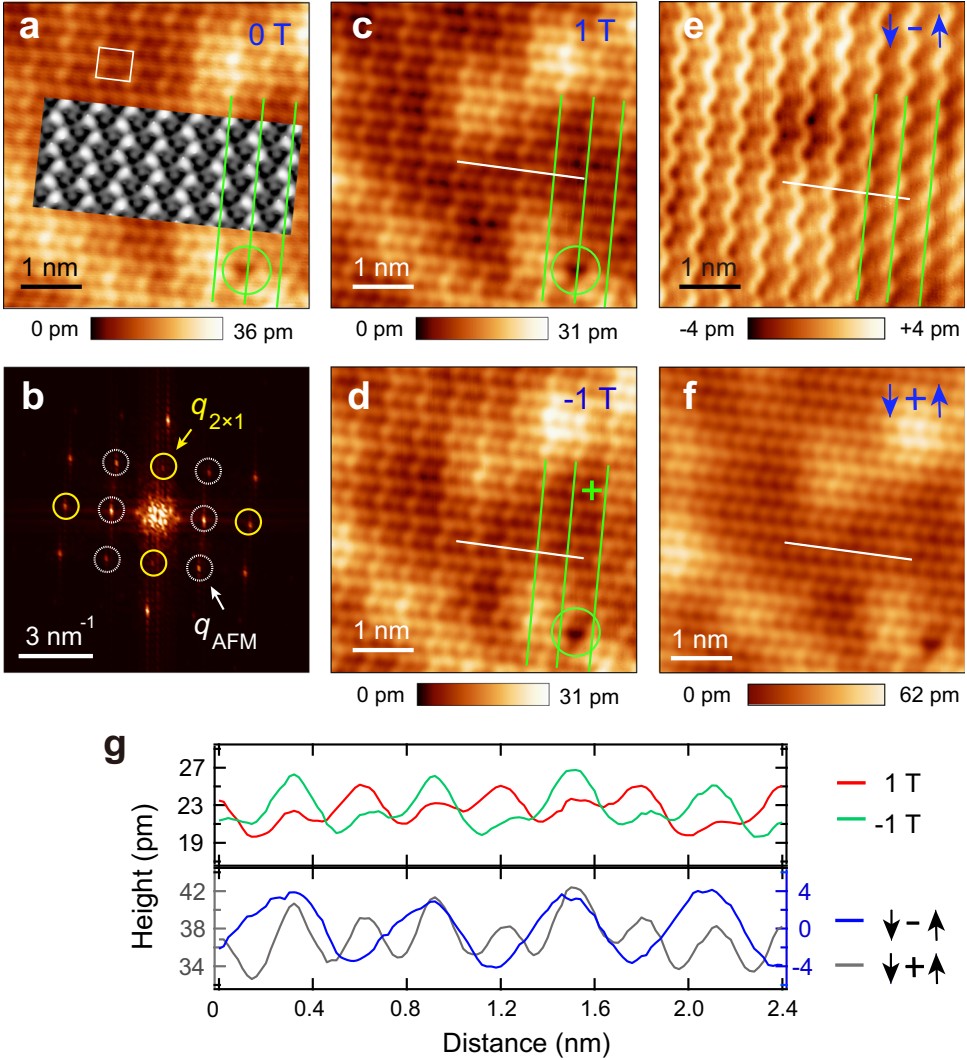

**Fig. 2 Spin mapping of the zigzag AFM spin texture in 1 ML 1T-CrTe₂. a**, **c**, and **d** SPSTM images ($V_b = -0.1$ V, $I_t = 100$ pA) of 1 ML CrTe₂ taken with a Cr tip under different magnetic fields. Inset of (**a**) is a DFT-simulated SPSTM image of the ML CrTe₂ in the zigzag type AFM order. **b** FFT image of (**a**). **f** Sum image of (**d**) and (**c**). **g** Line profiles extracted along the white lines in (**c–f**). The green lines in (**a**, **c–e**) mark the direction of the zigzag spin rows.

the experimentally observed 0.3 Å shrink. Many studies suggest that the substrate could apply an appreciably strong in-plane strain to the 2D layers and modify their in-plane magnetic order[24,25]. Here, we adopt a $10 \times 3\sqrt{3}$ CrTe₂/$16 \times 4\sqrt{3}$ bilayer graphene with a rectangular supercell (Supplementary Fig. S12) to model the substrate effect. This superlattice gives a lattice constant of 3.4/3.7/3.7 Å for CrTe₂, well consistent with the experimental values, and corresponds to substrate-induced compressive and tensile in-plane strains of 5% along $a_1$ and 3% along with $a_2$, respectively.

In this superlattice, the zigzag AFM ground state is still valid and is 71.0 meV/Cr more stable than the FM order. Our calculation indicates that the strengthened zigzag AFM order mainly originates from the in-plane strain applied by the substrate (Supplementary Table S1), instead of the interfacial charge transfer (Supplementary Fig. S13). Direct removal of the substrate, namely keeping the freestanding $10 \times 3\sqrt{3}$ ML in the $16 \times 4\sqrt{3}$ lattice, reveals a slightly larger energy difference of 76.8 meV/Cr between FM and zigzag AFM (Supplementary Table S1). The similarity of those two energy differences suggests a minor effect of the interfacial charge transfer. Given this energy difference of over 70 meV/Cr, the Néel temperature ($T_N$) of ML CrTe₂ was estimated at over 540 K using

the mean-field approximation, which is the upper limit of the transition temperature.

Spin densities are also mapped on the atomic structure of the CrTe₂ ML (Fig. 3a, b). Each Te atom is spin-polarized with a net magnetic moment of $-0.04\ \mu_B$ (Supplementary Table S1) by its three adjacent Cr atoms. Consequently, three $p$ orbitals of the Te atom are categorised into two groups, namely, $p_{x/y}$ and $p_z$ orbitals, with opposite spin polarisations (in green and red, respectively). In the SPSTM measurements, variation of spin-polarized tunnelling current is dominated by the Te-$p$ orbitals, because the current exponentially decays with the distance between the wavefunctions of the tip and the sample. The $p_z$ orbital has a dumbbell-like shape, pointing along with a Cr–Te bond. The $p_{x/y}$ orbital appears with a bagel-like shape, with its basal plane orthogonal to the $p_z$ dumbbell axis. A simulated spin-resolved image based on the zigzag AFM configuration (Fig. 3a, b) well reproduces those key features observed in the SPSTM (Fig. 2a) that each of those bright features presents the position of a surface Te atom and the $p_{x/y}$-like spin density shows higher intensity in the SPSTM image (Fig. 3b).

Figure 3d plots the angular distribution of magnetic anisotropy energy (MAE) in spherical coordinates. Angles $\theta$ and $\phi$ correspond to the angles between the magnetisation direction and the

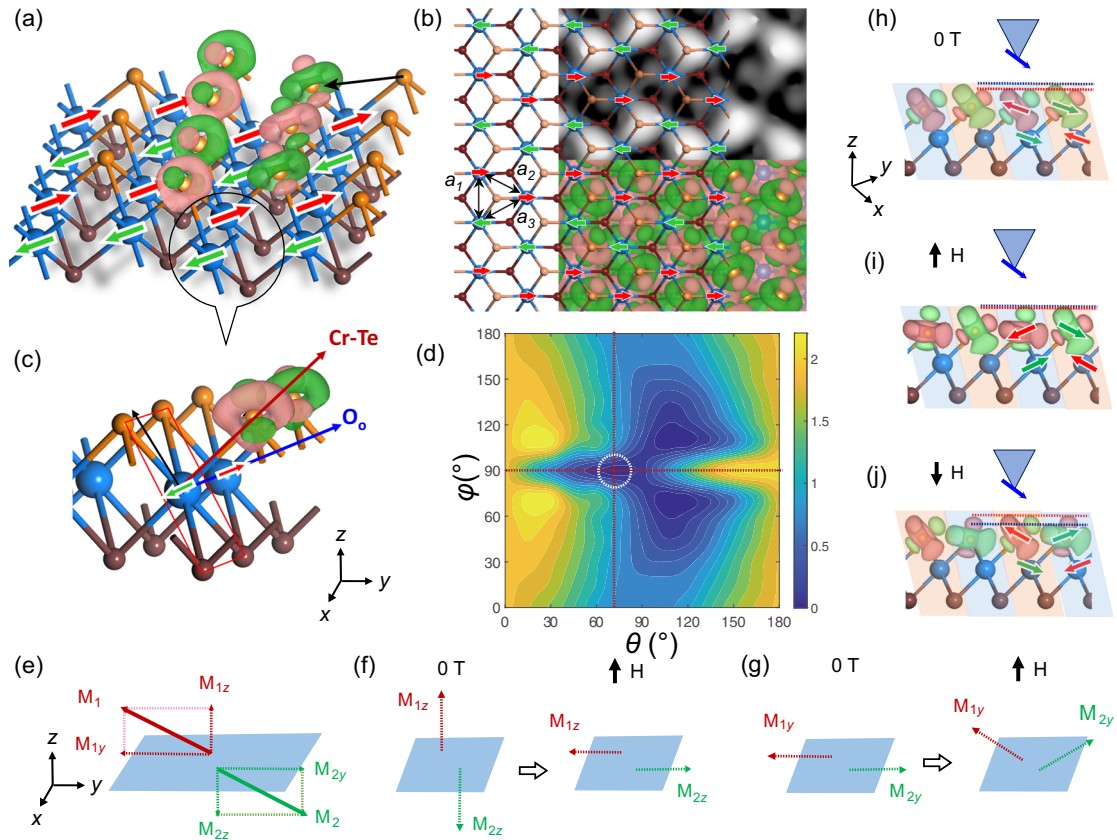

**Fig. 3 Zigzag AFM order and spin reorientation transition in ML CrTe₂. a** Perspective view of the CrTe₂ ML in the zigzag AFM order. Green and red arrows represent the magnetisation directions of Cr atoms. Spin density contours of the selected Te atoms are plotted with an isosurface value of 0.001 e/ Bohr³, where the red (green) contours denote the spin-up (spin-down). **b** Top view of the CrTe₂ ML in the zigzag order, superimposed on the DFT-simulated SPSTM image (top) and the spin density contours of the Te atoms (bottom). **c** Magnetisation axes in the MAE calculation. $x$, $y$ and $z$ axes correspond to the directions of the lattice vectors. A Cr-Te plane is marked with a red rectangular. **d** Angular dependence of the calculated MAE. Here, $\theta$ and $\phi$ correspond to the angles between the magnetisation direction and the $z$ and $x$ axes, respectively. A step size of 10° is used in our calculations. The total energy of the Cr moment oriented to the $O_o$ direction was chosen as the zero-energy reference. **e** Schematics showing magnetic moments, $M_1$ and $M_2$, of two adjacent Cr coupled antiferromagnetically, which are decomposed along the $z$ and $y$ axes (dashed arrows). **f, g** Evolution of the $z$-component (**f**) and $y$-component (**g**) of the magnetic moments under a vertical field. **h–j** Schematics of the SPSTM measurement configurations under different magnetic field directions. The ML CrTe₂ is viewed along the black arrow in (**a**). The tip magnetisation direction is represented with blue arrows. Magnetisation directions of the Cr moment and the Te-$p_{x/y}$ orbitals are marked.

$z$ and $x$ axes, respectively (Fig. 3c and Supplementary Fig. S14). The $O_o$ direction, which is in the $y$-$z$ plane and 70° off the $z$ axis, is the easy axis for magnetisation (Fig. 3c, d and Supplementary Tables S2, S3). The magnetisation direction along $y$ is only 0.12 meV/Cr less stable than that parallel to the easy axis. How-ever, it requires a 1.91 meV/Cr energy to rotate the magnetisation direction completely to $z$, making it less achievable under a small magnetic field.

The magnetic moments and their associated orbitals were theoretically found to undergo a spin flop-initialised canting process and rotations under an out-of-plane magnetic field, which appear a spin-contrast reversal of the SPSTM images, as explained below. The moment oriented in the $O_O$ direction is nearly perpendicular to the applied out-of-plane field. A spin-flop transition was expected to rotate the moments parallel to the $z$ direction. Given the lower limit of the exchange energy of $H_E = 76.8$ meV/Cr (Supplementary Table S1) and the MAE of $H_A = 1.91$ meV/Cr (Supplementary Table S3, energy difference between magnetisation along the $O_o$ and $z$ directions), it, how-ever, requires a spin-flop field of over 48 T according to the classical spin-flop picture. Including the substrate in our calcu-lations would slightly increase the spin-flop field to 49.1 T with $H_E = 71.0$ meV/Cr    and    $H_A = 2.16$ meV/Cr    (Supplementary

Tables S1, S2). This field is nearly two orders of magnitude larger than the observed threshold field of spin contrast reversal (~0.2 T), suggesting a different spin-flop picture[31].

As shown in Fig. 3e, we can decompose the magnetic moment of each Cr atom into two components, i.e., parallel to the $y$ and $z$ axes, denote M$z$ and M$y$, respectively, which do rotate or flop as a function of the magnetic field along $z$. We then discuss the field-dependent moment rotation using two adjacent and anti-parallel oriented moments, namely $M_1$ and $M_2$. In terms of $M_z$ (along $z$, Fig. 3f), a growing magnetic field parallel to the $z$ axis causes moments $M_{1z}$ and $M_{2z}$ flopping to the $y$ direction, which is the second easy axis with an MAE of 0.12 meV/Cr, corresponding to a threshold field of~ 0.35 T and consistent with the small experimental critical field of 0.2 T. Moments $M_{1y}$ and $M_{2y}$ gra-dually rotate to the $z$ direction with a growing magnetic field along $z$, thus leading to a noncollinear magnetic configuration that shows an emerging net magnetic moment along the $z$ direction (Fig. 3g). Response behaviours of the $z$- and $y$-compo-nents of the magnetic moments under a vertical magnetic field could be described using 'spin-flop' and 'canting' transitions, respectively. We thus term this field-dependent process as a spin reorientation transition. Our constrained noncollinear DFT cal-culation reveals that a 2° rotation of magnetic moment towards $z$

only requires a moderate energy cost of 0.15 meV/Cr, but the energy cost speedy increases for larger rotation angles, e.g., over 48 T for 90° (Supplementary Fig. S15). A significant decrease of the slope could be observed at smaller fields, which indicated a 'softer' potential near the anti-paralleled zigzag ground states.

Figure 3h–j schematically show the tip magnetisation and the spin configurations of the ML CrTe$_2$ under magnetic field, where the tip magnetisation is assumed parallel to the green arrow shown in Fig. 3h for simplicity. At 0 T, the green-coloured $p_{xy}$ orbitals are more extended to the tip position than the green-coloured $p_z$ orbital, as indicated using the blue and red dashed lines in Fig. 3h. The larger orbital extension thus leads to a larger tunnelling current, showing brighter spin contrast on the associated Te atoms, as highlighted using light-red and blue transparent parallelograms, respectively in Fig. 3h. Under a positive field, the Cr moments marked in green rotate anti-clockwise and those in red clockwise but with a smaller angle, producing a net magnetisation along the field direction (Fig. 3i). The magnetised Te-$p$ orbitals rotate concertedly with the Cr moments. This makes the magnetisation of the green-coloured $p_{xy}$ ($p_z$) orbital nearly parallel (orthogonal) to the tip magnetisation. The spin contrast consequently becomes more enhanced and maintains its phase relative to that of the 0 T. Under a negative field, a reversed spin rotation occurs to the Cr moments and the associated Te orbitals (Fig. 3j). The magnetisation of the green-coloured Te-$p_z$ ($p_{xy}$) orbital tends to be parallel (orthogonal) to the tip magnetisation, causing spin contrast reversal compared to 0 T. Around the spin-reversal field, the spin contrast is minimal, as is observed experimentally (Supplementary Fig. S8). While using a non-magnetic W tip, both the $p_{x/y}$ and $p_z$ orbitals of a Te atom could be detected simultaneously, which does not show spin resolution and has blurry orbital resolution. The eliminated or/and blurry resolutions thus lead to inappreciable changes of charge density between different spins or among different orbitals in STM images acquired using the W tip.

Given this unconventional spin reorientation transition picture established, we next compare the spectroscopic features from the STS and the DFT calculations. As shown in Fig. 4a, the density of states (DOS) of the ML CrTe$_2$ with the magnetic moments pointing to the $O_o$ direction (easy axis) appreciably differs from that to the $y$ direction (secondary easy axis), especially for the states residing from −0.35 to 0.25 eV. A ~10-meV upward shift is identified if the moments collinearly rotate from the easy to the secondary easy axes, which is, in principle, detectable by scanning tunnelling spectroscopy (STS). However, as illustrated in Fig. 3a–j, the moments rotate in a noncollinear manner, which causes mixing between the two spin channels and/or among different orbitals[32,33]. This noncollinear picture suggests that the major portion of orbitals that contribute to the tunnelling current, most likely, switches between the $p_z$ and $p_{x/y}$ orbitals under the $+z$ and $−z$ magnetic fields. We thus examine the DOSs projected on the $p_z$ and $p_{x/y}$ orbitals, as shown in Fig. 4b, to examine if they show apparent differences. A pronounced peak (P$_1$) sits at −0.44 eV, which is dominated by the $p_z$ and $p_{x/y}$ orbitals. Two additional peaks (P$_2$ and P$_3$) reside at −0.56 and −0.60 eV, respectively, the major compositions of which are $p_{x/y}$ orbital. The magnetic field-induced rotation of the Cr moment described earlier expects the variation of the $p_{x/y}$ and $p_z$ orbitals and their associated peaks in the contribution of the tunnelling conductance. Hence, a field-induced shift of ~150 meV, corresponding to the energy spacing between P$_1$ and P$_2$, is expected to be detectable in the spin-resolved spectra. Visualised wavefunction norms of the state P$_1$, depicted in Fig. 4c, d, show largely reduced $p_z$ orbitals and lightly lying-down Te-$p_{x/y}$ orbital if the moment rotates from the $O_o$ to $y$ directions. This finding further suggests the feasibility of the field-induced spin reorientation picture that leads to different SP-STS spectra.

We examine such an effect by taking spectra with the same micro Cr tip as that in Fig. 2 under various magnetic fields. The spin-resolved d$I$/d$V$ spectra (Fig. 4e) feature two characteristic peaks at approximately −0.5 and 0.88 V. Although the peak at 0.88 V stays unchanged, the peak at −0.5 V shifts progressively toward lower energy with increasing field along the $z$ orientations, by up to ~120 meV at 5 T. This value is similar with that of the calculated results. The field-induced peak shift is reproducible with a different Cr tip (Supplementary Fig. S16). Furthermore, the energy of the different Te-$p$ orbitals expects a spatial variation that correlates with the zigzag spin pattern, as is also unequivocally observed in the spin-resolved spectra (Supplementary Fig. S17). Therefore, such a spectral peak shift indeed supports the picture of the magnetic field-induced spin-mixing and reorientation of Te-$p$ orbitals.

Second-harmonic generation (SHG) and superconducting quantum interference device (SQUID) measurements were conducted ex-situ to confirm the observed AFM order and the field-induced spin reorientation transition. As shown in Supplementary Fig. S18, a significant SHG signal was observed at 7.6 K and gradually decreases with increasing temperature, until a tendency of saturation over 320 K. The SHG signal contrasts with the substrate, whose SHG signal is negligibly small and invariant with temperature. Since the inversion symmetry of the ML crystallographic structure is reserved, this temperature-dependent SHG signal would come from the broken time-reversal symmetry[34], i.e., an AFM order. While the saturated SHG signal over 320 K is, most likely, a result of the broken inversion symmetry at the CrTe$_2$/graphene interface, we cannot fully rule out the contribution from the AFM order. Nevertheless, this already suggests the AFM order survives at least up to 320 K.

The measured moment $m$ with SQUID monotonically increases from zero as a function of out-of-plane magnetic field $H$, which conforms to the AFM order (Supplementary Fig. S18). The increasing rate of $m$ becomes significantly reduced after the field reaches 0.5 T and the moment does not reach saturation under 5 T. All these features were well reproducible in our DFT simulations. According to our DFT results (Supplementary Fig. S15c), the net magnetisation along $z$ dramatically increases under a small magnetic field of no more than 0.8 T because of the emergence of the noncollinear order and the increasing rate is also reduced at higher magnetic fields. The $m$-$H$ curve suggests the AFM order sustains up to at least 300 K, consistent with that of the SHG measurement and the calculated upper limit of 540 K. All these observations strongly support the existence of the AFM order that were theoretically predicted and experimentally observed in SPSTM, as well as a spin-flop transition[35,36].

In conclusion, we observed a stable AFM order in ML CrTe$_2$ with SPSTM, whose spin contrast is from magnetically proximitized Te-$p$ orbitals with an easy axis in the $y$-$z$ plane and 70° off the $z$ axis. A spin reorientation transition occurs under magnetic fields, driving the ML CrTe$_2$ into a noncollinear spin texture. ML vdW systems can be readily accessed with external stimuli, such as electrostatic doping. As such, ML CrTe$_2$ provides a platform expecting versatile tunability for in-depth studies of fundamental physics in 2D antiferromagnetism, such as noncollinear spin-related magneto-transport phenomena[37], coherent many-body spin excitations at 2D limit[38], and layer-dependent magnetic order transition. This system also envisions to facilitate the development of AFM spintronics in miniaturised device applications.

## Methods

**Molecular-beam epitaxy (MBE) growth**. The graphene substrate was obtained via graphitisation of the SiC(0001) substrate with cycles of vacuum flashing treatment, the details of which are provided in[39]. The graphene obtained with such treatment is dominated by bilayers. The 1T-CrTe$_2$ films were grown on the graphene-covered

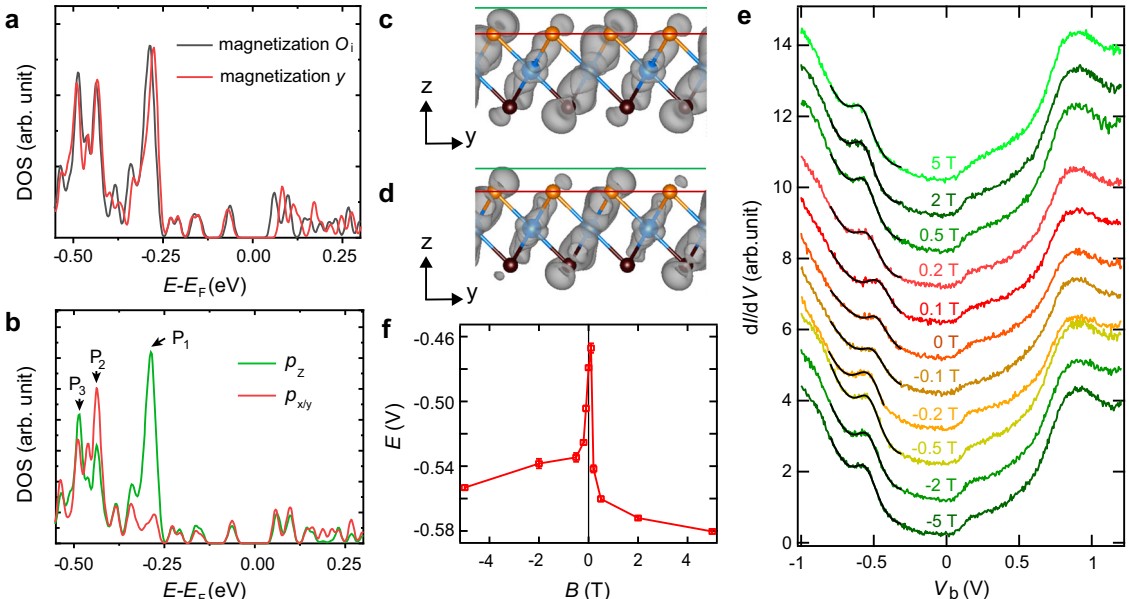

**Fig. 4 Electronic characteristics coupled with magnetism. a** DOS of the ML CrTe$_2$ supported on bilayer graphene, where the magnetic moment of each Cr cation is oriented along with the $O_o$ (the easy axis) and $y$ (secondary easy axis) directions. **b** DOSs projected on the Te-$p_z$ and $p_{x/y}$ orbitals in the ML CrTe$_2$ with Cr magnetic moments along the $O_o$ axis. Characteristic peaks $P_1$, $P_2$ and $P_3$ are marked. **c, d** Wavefunction norms of the $P_1$ peak with the Cr magnetic moments along with the $O_o$ (**c**) and $y$ (**d**) directions. The isosurface value of the contours is 0.0002 $e$/Bohr$^3$. Red and green lines indicate the positions of Te cations and the upper boundary of the contours, respectively. **e** Tunnelling spectra ($V_b = -1$ V, $I_t = 100$ pA, $V_{mod} = 14.14$ mV$_{rms}$) of the same location indicated in the green cross of Fig. 2d obtained with the same micro Cr tip at various magnetic fields. The black curves are fitted to the peaks around $-0.5$ eV with a Lorentz shape and a polynomial background. There is a conductance shoulder at about 0.1 V, which is also observable with a W tip (Supplementary Fig. S2). **f** Statistics of the fitted peak energies of (**e**) against the magnetic field, where the error bars are from the fitting.

SiC (0001) substrate via co-evaporation of high-purity Cr (purity, 99.995%) and Te (purity, 99.999%) atoms with a flux of ~1:30 according to MBE. The substrate was kept at 573 ± 10 K to facilitate the chemical formation of 1T-CrTe$_2$, where the substrate temperature was monitored with an infra-red spectrometer with an emissivity of 0.9. We found that the substrate temperature is crucial for growing single-phase CrTe$_2$ films. For ex-situ measurements, the CrTe$_2$ films were protected against degradation with Te capping layers of ~200 nm thickness.

**STM measurements**. The measurements were performed in a custom-made Unisoku STM system[40] mainly at 4.2 K unless described exclusively. The spin-averaged STM data were measured with an electrochemically etched W wire, which had been characterised on an Ag (111) surface prior to the measurements. The SPSTM data were taken with Cr or Fe tips. The Cr tip was prepared by coating ~5 layers of Cr (purity: 99.995%) on a W tip, which had been flashed to ~2000 K to remove oxides, followed by annealing at ~500 K for 10 min. The Fe tip was pre-pared by coating ~30 layers of Fe on a W tip, following similar flash and annealing procedures as the Cr tip. The tunnelling spectra were obtained through a lock-in detection of the tunnelling current with a modulation voltage of 983 Hz. The topographic images were processed by WSxM.

**STEM imaging**. The cross-section STEM specimens were prepared using a routine focused ion beam. STEM imaging, EDS and EELS analysis on the MBE-grown CrTe$_2$ were performed on a FEI Titan Themis with an extreme field emission gun and a DCOR aberration corrector operating at 300 kV. The inner and outer col-lection angles for the STEM images ($\beta_1$ and $\beta_2$) were 48 and 200 mrad, respectively. The convergence semi-angle of the probe was 25 mrad. The beam current was approximately 100 pA for the HAADF imaging, EDS and EELS chemical analyses. All imaging procedures were performed at room temperature.

**SHG measurements**. The SHG data of CrTe$_2$ and the substrate were from two different samples. The SHG was excited using a pulsed laser at the wavelength of 1200 nm with the pulse width of ~100 fs and the repetition rate of 80 MHz, and the signal wavelength of 600 nm was collected.

**SQUID measurements**. The magnetisation measurements of ML CrTe$_2$ and the substrate were carried out with a SQUID magnetometer in a physical property measurement system from Quantum Design. Magnetic fields were applied perpendicular to the sample surface.

**DFT calculations**. The calculations were performed using the generalised gradient approximation and projector augmented-wave method[41] as implemented in the Vienna ab-initio simulation package[42]. A uniform Monkhorst–Pack $k$ mesh of $15 \times 15 \times 1$ was adopted for integration over the Brillouin zone. An orthorhombic $2 \times 2\sqrt{3}$ supercell was used to show the zigzag AFM order in the ML CrTe$_2$, with a $k$ mesh of $10 \times 6 \times 1$. The magnetic anisotropic energy was derived using a $14 \times 8 \times 1$ $k$ mesh for reaching the convergence to 0.1 meV/Cr. A kinetic energy cutoff of 700 eV for the plane-wave basis set was used for the structural relaxation and electronic structure calculations. A sufficiently large vacuum layer over 20 Å along the out-of-plane direction was adopted to eliminate the interaction among layers. Dispersion correction was performed at the vdW-DF level[43], with the optB86b functional for the exchange potential[44], which was proven to be accurate in describing the structural properties of layered materials[45–47] and was adopted for structural-related calculations in this study. All atoms were allowed to relax until the residual force per atom was less than 0.01 eV/Å. To compare energy among different magnetic configurations, we used the PBE functional[48] with consideration of the spin–orbit coupling, based on the vdW-DF-optimised atomic structures. The on-site Coulomb interaction to the Cr $d$ orbitals had $U$ and $J$ values of 3.0 eV and 0.6 eV, respectively, as revealed by a linear response method[49] and comparison with the results of HSE06[50]. These values are comparable to those adopted in modelling CrI$_3$[51,52], CrS$_2$[27], and CrSe$_2$[28]. The role of $U$ values on the predicted magnetic ground state and the logics of choosing a proper $U$ value are discussed elsewhere. A $10 \times 3\sqrt{3}$ CrTe$_2$/$16 \times 4\sqrt{3}$ bilayer graphene with a rectan-gular supercell was adopted for the calculations of the magnetic ground state (Supplementary Table S1), MAE (Supplementary Table S2) and density of states (Fig. 4a, b) to include substrate effect.

## Data availability
The data that support the findings of this study are available from the corresponding author upon reasonable request.

## Code availability
The code that supports the findings of this study is available from the corresponding author upon reasonable request.

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

## Acknowledgements

This work is funded by the National Key Research and Development Program of China (Grant Nos. 2017YFA0403501, 2018YFE0202700, 2018YFA0307000), National Science Foundation of China (Grant Nos. 11874161, U20A6002, 11522431, 11774105, 21873033, 11622437, 61674171, 12104504 and 11974422), Strategic Priority Research Program of Chinese Academy of Sciences (Grant No. XDB30000000), Fundamental Research Funds for the Central Universities, China, and the Research Funds of Renmin University of China, and Guangdong International Science Collaboration Project (Grant No. 2019A050510001). C.W was supported by the China Postdoctoral Science Foundation (2021M693479). TEM characterisations received assistance from SUSTech Core Research Facilities and technical support from Pico Creative Centre, which is also duly supported by the Presidential Fund and Development and Reform Commission of Shenzhen Municipality. Calculations were performed at the Physics Lab of High-Performance Computing of Renmin University of China, Shanghai Supercomputer Centre.

## Author contributions

J.J.X., J.H.N. and R.L. grew the sample and did the SPSTM experiments with the help of Z.Y.L., W.H.Z., Z.M.Z. and M.P.M.; M.H. and J.L. carried out the STEM experiments; Y.Y. and S.W. performed the SHG measurements; X.C., R.L., J.H. and Z.X. did the SQUID measurements; C.W. and W.J. performed the calculations; Y.S.F., W.J., J.J.X., C.W., J.H.N. and R.L. analysed the data, and wrote the manuscript, with comments from all authors. Y.S.F. and W.J. supervised the project.

## Competing interests

The authors declare no competing interests.
