## [Peer Review File · Nature Communications]

Reviewers' Comments:

Reviewer #1:

Remarks to the Author:

Review of NCOMMS-20-46354-T

Spin mapping of intralayer antiferromagnetism and spin-flop transition in monolayer CrTe₂ by J.-J. Xian et al.

The authors present a study of the magnetic structure of a monolayer (few layers) 1T-CrTe₂ grown on bilayer graphene on SiC utilizing spin-polarized scanning tunneling microscopy (SP-STM) in magnetic fields in combination with density functional theory (DFT). Additionally, STEM is used to characterize the growth of the films. They observe a zigzag-type pattern in SP-STM measurements that reverses its contrast in opposite magnetic field directions. Based on DFT calculations, they suggest an antiferromagnetic zigzag-type ground state and a spin-flop model to explain the field-dependent spin-contrast reversal. The authors suggest that their work "adds a new dimension for understanding intralayer magnetic order at the 2D limit."

It is of no doubt that the magnetism in magnetic 2D materials is a hot topic as these material class may be the basis for novel nanoscale device applications. So far, atomic-scale magnetic detection remains challenging, and not a lot experimental work has been able to address the magnetic properties at the atomic scale owing to the challenging preparation of the 2D films suited for atomic-scale experiments. I give highest credit to the authors to combine the challenging experimental growth and characterization methods, along with first principles calculations, which is a highly suited approach. Despite a few language flaws, the manuscript is written in a manner that is easy to follow. Although find the work of high interest and potentially high impact, I have a few serious doubts about the manuscript. After having read the manuscript, I am left with the impression that the authors oversell their findings. In addition, I am missing a few experimental details or/and I do not agree with some conclusions. I detail my issues in the list below. Based on this list, I cannot recommend publication in Nature Communication.

Major points:

- In the conclusion/outlook they write about "a new dimension", and that this system is suited for in-depth studies for, e.g. spin excitation. I do not understand what this new dimension is. In addition, what type of spin excitation do the authors think of? STM is actually a tool to probe spin excitations at surfaces, have they observed any signal?
- Why did they choose for CrTe₂ for their study? Why is this specific material a good representative of magnetic 2D materials? Can their findings be generalized for other 2D materials?
- In the manuscript, the authors write that they suggest a spin-flop model which is reasonable. In the conclusion they state that "a spin-flop transition occurs". I find this statement a bit too strong given the discussion of evidences. I detail a few points in the following:
 - The authors observe a threshold magnetic field in their experiments of $< 0.2T$ for the contrast reversal. How does this value compare with a prediction from their theoretic model?
 - Line 229: The authors state the peak at $-0.5V$ shifts progressively toward lower energy with increasing field. However, if I look at Fig. 4f, the E vs. B plot of the peak position shows a quite complex behavior with a peak/maximum, and not a monotonous behavior (which I would expect). I do not understand this at all. What is the reason for that?
 - What about the shoulder at $0.1V$ in Fig. 1e? What is that?
 - The authors state that the data in Fig. S12 taken with a different Cr tip reproduces their data. I disagree. What I see is that in Fig. 4f: $E(1T) < E(0T) < E(-1T)$, and in Fig. S12: $E(1T) < E(-1T) < E(0T)$. Can they please comment on that?
- In line 84 they write about the distinct electronic structure between the graphene and CrTe₂. How do they know? They do not provide any $dIdV(V)$ data with a spin-averaged tip, which is data that can be easily obtained in their setup. I am also missing this piece of data to correctly interpret Figure S12 and S13 (so the spin-resolved $dIdV$ spectra). I strongly encourage the authors to add

this detail.

- It is not clear to me that the dark blue colored layer in Fig. 1b is graphene. Fig. 1b does not show atomic resolution that could be used as an evidence, nor does Fig. 1d indicate for me why there is a graphene layer. For me, the assigned graphene layer appears similar to a few of the SiC layers. I ask the authors to make this statement clearer and give more evidence, or/and explain it better.
- Line 92: They write about the interlayer vdW gap that should be seeable in Fig 1d. It is not clear for me where it is. I suggest to improve the contrast of Fig. 1d and add more explanation.
- The authors define three lattice constants, a_1 , a_2 , and a_3 , and claim that these deviate from rectangular ones, which would be in line with the DFT calculations. However, these are rather small deviations, and I am missing a discussion about the error margin for determining the lattice constants from the FFT image, as well as an influence of different scanner calibrations/piezo scanner asymmetry in the x and y direction.
- Line 119: The authors write that the tip magnetization of the Cr tip is canted (also in line 146). How do they know that? Figure S7 (Cr tip) does not convince me as for me the contrast appears the same on the different domains. I agree that in Figure S8, for the Fe tip, there seems to be an in-plane component of the tip magnetization.
- Line 141: The statement "memorizes the history of the field" I cannot follow by looking at Fig. S6. Can the authors lay this out more?
- Figure S13 shows various bright spots on the surface. What are they? How can they be sure that these spots do not modify the position of the peak around -0.5V?

Minor points:

- Line 108: they talk about the 6x6 Moiré pattern at the graphene/SiC interface. I am missing a citation here.
- Figure S3 looks as if its processed/filtered. Can the authors comment on that?
- The zoom in Fig. 1f shadows a lot of the large-scale picture which is unfortunate. Can the author find a better way of presenting it?
- Line 167: The authors indicate that in-plane strain plays a significant role as they infer from their calculation. Can the authors give more details about the order of magnitude of the strain?
- The way how Fig. S11 is plotted suggests that the energy for the sAFM-AABB does not depend on the doping. Is this correct?

Reviewer #2:

Remarks to the Author:

This work reports the study of monolayer CrTe₂. Main claims include: (1) growth of monolayer CrTe₂; (2) CrTe₂ has zigzag antiferromagnetic order with easy axis along the Cr-Te bonding direction; (3) observation of spin-flop type transition in real space. Bulk crystal CrTe₂ has been demonstrated as ferromagnet with high Curie temperature. The reported results can be potentially very interesting if the claims are correct. Unfortunately, the current status of the paper is a bit premature and I cannot recommend it for publication.

(1) I am not convinced that there is even a magnetic order. Temperature dependence could provide useful information of magnetic phase transition. For example, how does the observed zigzag pattern depend on temperature? Ideally, the authors need to have a second approach to demonstrate the formation of magnetic order.

(2) The growth produces both monolayer and bilayer. What do you see in bilayer? Is the zigzag

pattern special to monolayer?

(3) What is the magnetic response of Cr tip? In tunneling measurements, the spin structure of Cr will affect the characterization of CrTe₂. How do you characterize the Cr tip magnetization orientation? How do you experimentally conclude the spin orientation in CrTe₂?

(4) What is the saturation field along the z direction? If the magnetic field (e.g. 5T) is larger than the saturation field, then all spins should be aligned in the Z direction. Then I do not expect to see a zigzag pattern under large magnetic field, as those in Fig. S5. In addition, at B field of 5T, the spin orientation of Cr tip will change as well. But the zigzag patterns in Fig. S5 does not change at all. This is confusing.

(5) The authors observed domain effects (Fig. S7). Could the structural domain give rise to the observed domain pattern?

Reviewer #3:

Remarks to the Author:

The authors present a study on CrTe₂ monolayer in which they use spin polarized scanning microscopy to conclude that the material is an antiferromagnet, and to probe a spin-flop transition that occurs in the presence of an applied magnetic field.

I have a number of substantial concerns about the work. Most of these concerns stem from what seems to be a lack of sufficient control experiments, or other types of sample interrogation which could corroborate the antiferromagnet order suggested by the authors.

1. There should be control experiments provided on the tip which demonstrate that the tip magnetization is unaffected by the magnetic field. This seems to be the most obvious potential artifact, and it is simply asserted in the text that the tip is impervious to magnetic fields up to 2 T (without reference to previous work or to experimental data) (line 145). Also, the additional experiment using an Fe tip is used as a control. The text seems to imply that the stability of the Fe tip is more established, although I'm not sure. However, again no reference is provided. This seems quite unexpected to me, and it is crucial to conclusively establish that the tip is not introducing magnetic field-dependent artifacts. This concern is amplified given some of the aspects of the data, as described next.

2. The magnetic field-dependence does not seem consistent with a spin-flop transition. A spin-flop transition typically occurs at a nonzero threshold field, given by $(2 * H_{\text{exchange}} * H_{\text{anisotropy}})^{1/2}$. Using the density functional theory estimates of the anisotropy energy (on the order of 1 meV/Cr atom), and exchange energy (I come up with 60 meV/Cr atom as the energy cost of flipping one Cr atom direction in the staggered configuration), I estimate the spin-flop field is 55 T! This is a huge field, although the way I estimate exchange energy based on your DFT result is quite crude. (Also, as an aside, the DFT calculation of magnetocrystalline anisotropy can be very delicate, requiring very fine k-mesh. The authors should check that their results are well converged - I estimate the anisotropy field computed as 5 T, which is quite large.) Regardless of the value inferred from DFT, the experimental data seems to show that the signal ascribed to spin-flop evolves smoothly from zero field. In other words, there's no threshold field, and there should be. There should be. Returning to numbers: for CrI₃, $H_{\text{anisotropy}} = 1.7$ T and $H_{\text{exchange}} = 0.76$ T, resulting in spin-flop field of 1.6 T (<https://arxiv.org/abs/2001.04044>).

3. Given all of the above, it would be quite worthwhile to perform additional experiments on this material to find other evidence of spin-flop transition. For example, you can do XMCD, or squid measurements to detect the formation of magnetization associated with this state (as in <https://journals.aps.org/prb/pdf/10.1103/PhysRevB.101.094429>). Also, measurements to estimate the Neel temperature would be useful, depending on how conclusive things look after additional control measurements described above.

Reply to review

Reviewer #1 (Remarks to the Author):

Review of NCOMMS-20-46354-T

Spin mapping of intralayer antiferromagnetism and spin–flop transition in monolayer CrTe₂ by J.-J. Xian et al.

The authors present a study of the magnetic structure of a monolayer (few layers) 1T-CrTe₂ grown on bilayer graphene on SiC utilizing spin-polarized scanning tunneling microscopy (SP-STM) in magnetic fields in combination with density functional theory (DFT). Additionally, STEM is used to characterize the growth of the films. They observe a zigzag-type pattern in SP-STM measurements that reverses its contrast in opposite magnetic field directions. Based on DFT calculations, they suggest an antiferromagnetic zigzag-type ground state and a spin-flop model to explain the field-dependent spin-contrast reversal. The authors suggest that their work “adds a new dimension for understanding intralayer magnetic order at the 2D limit.”

It is of no doubt that the magnetism in magnetic 2D materials is a hot topic as these material class may be the basis for novel nanoscale device applications. So far, atomic-scale magnetic detection remains challenging, and not a lot experimental work has been able to address the magnetic properties at the atomic scale owing to the challenging preparation of the 2D films suited for atomic-scale experiments. I give highest credit to the authors to combine the challenging experimental growth and characterization methods, along with first principles calculations, which is a highly suited approach. Despite a few language flaws, the manuscript is written in a manner that is easy to follow. Although find the work of high interest and potentially high impact, I have a few serious doubts about the manuscript. After having read the manuscript, I am left with the impression that the authors oversell their findings. In addition, I am missing a few experimental details or/and I do not agree with some conclusions. I detail my issues in the list below. Based on this list, I cannot recommend publication in Nature Communication.

We thank the reviewer for his/her recommendation and comments that highlight the importance, general interest and experimental challenges of our work. We acknowledge the reviewer for raising the insightful critics/concerns over the unclear issues of our manuscript, which certainly help with improving our work. Our point-to-point responses to the reviewer’s concerns are as follows.

Major points:

Q1 In the conclusion/outlook they write about “a new dimension”, and that this system is suited for in-depth studies for, e.g. spin excitation. I do not understand what this new dimension is. In addition, what type of spin excitation do the authors think of? STM is actually a tool to probe spin excitations at surfaces, have they observed any signal?

We apologize for not describing this term clearly. Recently, coherent many-body spin excitations have been found in bulk van der Waals antiferromagnet NiPS₃ (Nature 583, 785–789(2020), Ref. 37). In the CrTe₂ monolayer, its intrinsic antiferromagnetic order, which may be readily manipulated with external stimuli such as electrostatic doping, provides a platform for studying such coherent many body excitations at the 2D limit. We agree with the reviewer that spin excitations, in certain cases, can be detected with scanning tunneling spectroscopy. In our experiment, a spectral dip shown in Fig. R2(b) was observed around the Fermi level, which is possibly related to spin excitation. Further in-depth studies are on-going to understand such spectral features.

For clarity, we have revised the sentence about “a new dimension” to “ML vdW systems can be readily accessed with external stimuli, such as electrostatic doping. As such, ML CrTe₂ provides a platform expecting versatile tunability for in-depth studies of fundamental physics in 2D antiferromagnetism, such as noncollinear spin-related magneto-transport phenomena, coherent many body spin excitations at 2D limit, and layer-dependent magnetic order transition.”.

Q2 Why did they choose for CrTe₂ for their study? Why is this specific material a good representative of magnetic 2D materials? Can their findings be generalized for other 2D materials?

Thanks for raising this important issue. We choose CrTe₂ for our study from the following considerations. First, bulk CrTe₂ is a strong ferromagnet. In view of the seminal work of monolayer CrI₃ and Cr₂Ge₂Te₆, this raises the possibility of obtaining high temperature monolayer ferromagnet with CrTe₂. Second, theory predicts a distance dependent interlayer magnetic phase transition in CrTe₂ few layers, due to competing interactions between Pauli and Coulomb repulsions and kinetic-energy gain (Ref. Phys. Rev. B 102, 020402 (2020)). It is thus interesting to motivate the experiments to study its magnetic order at monolayer limit.

This finding is envisioned to be generalized to other 2D materials. Few layer CrS₂ and CrSe₂ may share very similar magnetic properties, e.g. in-plane AFM in the monolayer and in-plane FM in the bulk counterpart, at least from the view of theoretical calculations (Ref. Phys. Rev. B 97, 245409 (2018) and Phys. Rev. B 102, 020402 (2020)). While our theory predicted that such transition should be shared with CrTe₂, none of these three monolayers was experimentally observed. Therefore, those systems are on the list of our immediate next investigations. Other 2D materials are also waiting to be explored.

For clarity, we have moved a paragraph from Page 6 in our original version to Page 4, to explain the reasons for choosing CrTe₂ in our study.

Q3A In the manuscript, the authors write that they suggest a spin-flop model which is reasonable. In the conclusion they state that “a spin-flop transition occurs”. I find this statement a bit too strong given the discussion of evidences. I detail a few points in the following:

- The authors observe a threshold magnetic field in their experiments of < 0.2T for the

contrast reversal. How does this value compare with a prediction from their theoretic model?

As shown in Fig. R1a, the O_o direction, in the y - z plane and 70 degrees off the z axis, is the easy axis direction of ML CrTe₂. The total moment is thus decomposable into two components parallel to the y and z axes, denote M_z and M_y , respectively, which do rotate or flop as a function of the magnetic field along z . We then discuss the field-dependent moment rotation using two adjacent and anti-parallel oriented moments, namely M_1 and M_2 . In terms of M_z (along z , Fig. R1c), a growing magnetic field parallel to the z axis causes moments M_{1z} and M_{2z} flopping to the y direction, which is the second easy axis with a MAE of 0.12 meV/Cr, corresponding to a threshold field of ~ 0.35 T and consistent with the small experimental critical field of ~ 0.2 T. For M_y , moments M_{1y} and M_{2y} gradually rotate to the z direction with a growing magnetic field along z , thus leading to a non-collinear magnetic configure that shows a net magnetic moment along the z direction (Fig. R1d). Our non-collinear constrained DFT calculation reveals that a 2-degree rotation of magnetic moment towards z only requires a moderate energy cost of 0.15 meV/Cr (Supplementary Figure S13a). However, complete rotation of the moment to the z direction needs a much larger energy to surmount the magnetic anisotropic energy of 1.91 meV/Cr (energy difference between magnetization along the O_o and z direction) and the intralayer spin-exchange energy of 76.8 meV/Cr; this requires a magnetic field over 48 T given by $\sqrt{2H_{exchange} \cdot H_{anisotropy}}$ according to the classical spin-flop picture, consistent with our observed unsaturated magnetization under a few Tesla.

To clarify these points, we revised relevant sentences on page 11 and Figure 3 of the main text for clarity.

Figure R1. Illustration of the rotation of magnetic moments under out-of-plane field. (a) Magnetization axes considered in the MAE calculation. Here blue dashed arrow corresponds to the easy axis O_o . (b) The green and red arrows indicate the magnetic moments M_1 and M_2 of Cr atoms of zigzag chain with different magnetization directions. The dashed arrows correspond to the decomposed magnetic moments component along z and y axis. (c-d) Evolution of the z -component (c) and y -component (d) of magnetic moments under vertical field.

Q3B Line 229: The authors state the peak at -0.5V shifts progressively toward lower energy with increasing field. However, if I look at Fig. 4f, the E vs. B plot of the peak position shows a quite complex behavior with a peak/maximum, and not a monotonous behavior (which I would expect). I do not understand this at all. What is the reason for that?

Thanks for raising this issue. Above 0.2 T, E vs. B is monotonous for the *same* field direction. In Fig. 4f, for the positive B , E monotonically shifts to negative energy with increasing B magnitude. The same monotonic behavior is observed for the negative B . Here, the positive (negative) B means the magnetic field is applied up (down) to the CrTe₂ basal plane. As a result, there apparently appears a peak around zero B in Fig. 4f. We apologize for leaving the impression that the E vs. B has a complex behavior.

For clarity, we have marked the zero B with a vertical line in Fig. 4f.

Q3C What about the shoulder at 0.1V in Fig. 1e? What is that?

The referee probably means Fig. 4e. The shoulder is also observed with a spin-averaged W tip. It is likely associated with a band extreme of CrTe₂.

For clarity, we have added a phrase "There is a conductance shoulder at about 0.1 V, that is also observable with a W tip." to the figure caption of Fig. 4.

Q3D The authors state that the data in Fig. S12 taken with a different Cr tip reproduces their data. I disagree. What I see is that in Fig. 4f: $E(1T) < E(0T) < E(-1T)$, and in Fig. S12: $E(1T) < E(-1T) < E(0T)$. Can they please comment on that?

Thanks for pointing out this issue. Fig. 4f and Fig. S12 (now Fig. S15) both have the relation $E(1T) < E(-1T) < E(0T)$. For clarity, we have given the fitted values of E as the following:

Fig. 4f: $E(1T) = -564.22$ meV, $E(-1T) = -535.93$ meV, $E(0T) = -479.22$ meV,

Fig. S12 (now Fig. S15): $E(1T) = -388.56$ meV, $E(-1T) = -384.29$ meV, $E(0T) = -290.94$ meV.

For clarity, we have indicated those numbers in the figure caption of updated Fig. S15.

Q4 In line 84 they write about the distinct electronic structure between the graphene and CrTe₂. How do they know? They do not provide any $dI/dV(V)$ data with a spin-averaged tip, which is data that can be easily obtained in their setup. I am also missing this piece of data to correctly interpret Figure S12 and S13 (so the spin-resolved dI/dV spectra). I strongly encourage the authors to add this detail.

Thanks for raising this important issue and the nice suggestion. We have added the spectra of graphene and CrTe₂ acquired with a spin-average W tip, as is seen in Fig. R2(a). This figure is added as Fig. S2 in supplementary material.

Figs. R2 Spin-averaged tunneling spectra of ML CrTe₂ and graphene. (a) Large energy scale spectra ($V_b = 1.5$ V, $I_t = 100$ pA, $V_{\text{mod}} = 14.14$ mV_{rms}) of ML CrTe₂ and graphene measured with a W tip. (b) Small energy scale spectrum ($V_b = 0.1$ V, $I_t = 100$ pA, $V_{\text{mod}} = 3.536$ mV_{rms}) of ML CrTe₂ around the Fermi level.

Q5 It is not clear to me that the dark blue colored layer in Fig. 1b is graphene. Fig. 1b does not show atomic resolution that could be used as an evidence, nor does Fig. 1d indicate for me why there is a graphene layer. For me, the assigned graphene layer appears similar to a few of the SiC layers. I ask the authors to make this statement clearer and give more evidence, or/and explain it better.

Thanks for the suggestion. For Fig.1(b), the evidence of graphene layer can be given from both its atomic resolution STM image and its dI/dV spectrum. Its dI/dV spectrum has been shown in Fig. R2(a). Its atomic resolution image is shown in Fig. R3(a). Furthermore, Figure 1d indeed shows a graphene layer between the CrTe₂ and SiC layers, as is seen from its weaker contrast and larger interlayer spacing than the underneath SiC lattice. To give further evidence, we add a new panel of C-K EDS mapping in the revised Fig. S3(e), in which the carbon signal dominates in the graphene layer, in a sharp contrast to the underneath SiC parts. Therefore, from both the dI/dV spectrum, the atomic STM image and the atomic EDS mapping of the cross-section, the graphene layer can be clearly identified between the SiC and CrTe₂.

We have added the following sentence in Page 5 of the main text and the new panel (e) in Fig. S3 to clarify this issue:

“The associated integrated differential phase contrast (iDPC) image (Fig. 1(d)) resolves the fine details of the film, clearly showing the CrTe₂ layer and the supporting graphene/SiC substrate, where the graphene layer is distinguished by its weaker contrast and larger interlayer spacing than the underneath SiC lattice.”

Q6 Line 92: They write about the interlayer vdW gap that should be seeable in Fig 1d. It is not clear for me where it is. I suggest to improve the contrast of Fig. 1d and add more explanation.

We revised the Fig. 1 (d) to better illustrate the vdW gap between the CrTe₂ layers. The negligible contrast between the two CrTe₂ layers in the revised Fig. 1(d) demonstrates clearly the interlayer vdW gap without intercalated Cr atoms. The vdW gap between graphene and CrTe₂ is also clearly seen, which is larger than the interlayer vdW gap in CrTe₂.

We added the following sentences in Page 5 of the revised manuscript:
“The negligible contrast between two CrTe₂ layers in Fig. 1(d) demonstrates clearly a vdW gap without intercalated Cr atoms. The vdW gap between graphene and CrTe₂ is also clearly seen, which is larger than the interlayer vdW gap in CrTe₂.”

Q7 The authors define three lattice constants, a_1 , a_2 , and a_3 , and claim that these deviate from rectangular ones, which would be in line with the DFT calculations. However, these are rather small deviations, and I am missing a discussion about the error margin for determining the lattice constants from the FFT image, as well as an influence of different scanner calibrations/piezo scanner asymmetry in the x and y direction.

We agree that the deviations are small. To evaluate the deviations, we provide atomic resolution of graphene shown in Figs. R3 (a) and (b), which is applied for calibrating the scanner. The imaged graphene lattices are measured as $a_1 = 2.46 \text{ \AA}$, $a_2 = 2.45 \text{ \AA}$, $a_3 = 2.46 \text{ \AA}$, in good agreement with the theoretical value of 2.458 \AA . Thus, the scanner has a negligibly small asymmetry of 0.4%, which is significantly smaller than that of the measured CrTe₂ lattice (~8%). A more direct evidence for the lattice deviation is the imaging of a CrTe₂ monolayer with a domain boundary. As is seen in Figs. R3(c) and (d), its FFT pattern evidently contains two sets. The FFT pattern of each individual domain changes to one set, as is shown in Figs. R3(e) and (f). The lattice constants of ML CrTe₂ determined on domain of (e) are $a_1 = 3.41 \text{ \AA}$, $a_2 = 3.72 \text{ \AA}$, $a_3 = 3.73 \text{ \AA}$. They are similar to those of (f), namely, $a_1 = 3.45 \text{ \AA}$, $a_2 = 3.71 \text{ \AA}$, $a_3 = 3.68 \text{ \AA}$. Since the two domains are imaged with the exactly same condition, the lattice distortions of CrTe₂ are unambiguously confirmed.

For clarity, we have revised Fig. S4 to reflect the above discussions.

Fig. R3 Atomic resolution of graphene and monolayer CrTe₂. (a,b) STM image ($V = -100$ mV, $I = 500$ pA) showing the atomic resolution of graphene (a) and its FFT image (b). The lattice units of graphene are indicated with a green triangle in (a). (c,d) STM image ($V = -500$ mV, $I = 10$ pA) showing the atomic resolution of monolayer CrTe₂ with a domain boundary (c) and its FFT image (d). (e,f) FFT images of the rectangle areas in (c).

Q8 Line 119: The authors write that the tip magnetization of the Cr tip is canted (also in line 146). How do they know that? Figure S7 (Cr tip) does not convince me as for me the contrast appears the same on the different domains. I agree that in Figure S8, for the Fe tip, there seems to be an in-plane component of the tip magnetization.

Thanks for raising this issue. The canted tip magnetization means the tip has both out-of-plane and in-plane magnetization. From Fig. 2, the spin-contrast responses to out-of-plane magnetic fields, demonstrating the Cr tip should have out-of-plane spin sensitivity. Fig. S7 (Fig. S8 in the revised version) shows an image of the different domains with the same Cr tip at zero magnetic field. If the Cr tip also has in-plane spin sensitivity, the spin contrast of different domains should vary.

To more clearly highlight such variations in spin contrast, we have replotted Fig. S7(now Fig. S8), which is displayed as Fig. R4 below. In the revised Fig. S7 (now Fig. S8), different spin contrast can be seen at different domains, which is similar to Fig. S8a (now Fig. S9a).

Fig. R4 SP-STM image of ML CrTe₂ with a Cr tip, showing different spin contrast. (a) SP-STM images ($V = 0.2$ V, $I = 100$ pA) of ML CrTe₂, which has a domain boundary. (b,c) Zoom-in images of (a), which are extracted from the left (b) and right (c) domain. Note that the images of (b) and (c) have been rotated for comparison. The blue rectangles mark the AFM unit cell of the ML CrTe₂. The spin-contrast of (b) is zigzag-like, while that of (c) is more rectangular-like.

Q9 Line 141: The statement “memorizes the history of the field” I cannot follow by looking at Fig. S6. Can the authors lay this out more?

In Fig. S6 (now Fig. S7), the magnetic field is applied following a sequence of $1T \rightarrow 0T \rightarrow -1T \rightarrow 0T$. For the spin contrast imaged at $0T$ of Fig. S6b and S6d, their spin contrast changes its phase, as is seen from the defect marker in white circle. This reflects the spin contrast is related to the history of the applied magnetic field.

For clarity, we have added a sentence “For the spin contrast imaged at $0T$ of (b) and (c), their spin contrast changes its phase, as is seen from the defect marker in white circle. This reflects the spin contrast is related to the history of the applied magnetic field.” to the figure caption.

Q10 Figure S13 shows various bright spots on the surface. What are they? How can they be sure that these spots do not modify the position of the peak around $-0.5V$?

The bright spots are from the 6×6 reconstruction between the graphene and SiC interface. They do not affect the peak around -0.5 V. The spectra of Fig. S13 (now Fig. S16) were acquired along two lines perpendicular to the zigzag spin stripes, and have different spectroscopic locations relevant to the bright spots. The extracted energies of the $-0.5V$ peak show regular modulations that correlate with the zigzag stripes, but irrelevant to the locations of the bright spots.

For clarity, we have added two sentences “There are some bright spots in (a), which are from the 6×6 reconstruction between the graphene and SiC interface. They do not affect the peak around -0.5 V, as is seen in (c).” in the figure caption.

Minor points:

Q11 Line 108: they talk about the 6x6 Moiré pattern at the graphene/SiC interface. I am missing a citation here.

Thanks for the suggestion. We have added the following reference as Ref. 30 in the main text.

Emtsev, K. V., Speck, F., Seyller, Th., Ley, L. and Riley, J.D. Interaction, growth, and ordering of epitaxial graphene on SiC(0001) surfaces: A comparative photoelectron spectroscopy study. *Phys. Rev. B* 77, 155303 (2008).

Q12 Figure S3 looks as if its processed/filtered. Can the authors comment on that?

Fig. S3 (now Fig. S4) is the raw data without any processing or filtering. For clarity, we have added a sentence “The images in (a) and (c) are the raw data without filtering.” in the figure caption.

Q13 The zoom in Fig. 1f shadows a lot of the large-scale picture which is unfortunate. Can the author find a better way of presenting it?

Thanks for the suggestion. We have revised Fig. 1f to reduce the shadowed area.

Q14 Line 167: The authors indicate that in-plane strain plays a significant role as they infer from their calculation. Can the authors give more details about the order of magnitude of the strain?

We appreciate the reviewer’s concern and regret with our not-clear-enough discussion on substrate induced strain effects. In our revision, we discussed in details the surface induced strains and considered the role of varying the on-site U value. In short, a substrate induced in-plane compressive strain of 5% along a_1 and tensile strain of 3% along a_2 was confirmed by our updated experimental and theoretical results.

Our U - J values used in the original manuscript was derived using a linear response method. It usually gives the upper limit of U values which may often overestimate lattice constants and sometimes fail to find the interlayer magnetic ground state of CrTe_2 (*Phys. Rev. B* 102, 020402 (2020)). A smaller U value, i.e. 3.0 eV, was considered, which is fitted to the HSE result of the distance-dependent energy different between interlayer FM and AFM configurations. Both U values show qualitatively consistent in terms of in-plane strains as we presented as follows.

In Q7, we refined our experimental lattice constants to $a_1= 3.4 \text{ \AA}$, $a_2= 3.7 \text{ \AA}$, $a_3 = 3.7 \text{ \AA}$ using both real- and reciprocal-spaces techniques. We compared this slightly updated experimental value with our theoretical values with $U=4.4$ and 3.0 eV. The smaller U value yields the fully relaxed free-standing CrTe_2 lattice as $a_1= 3.6 \text{ \AA}$, $a_2= 3.6 \text{ \AA}$, $a_3 = 3.7 \text{ \AA}$, suggesting a substrate induced in-plane compressive strain of 5% along a_1 and tensile strain of 3% along a_2 , which could be modeled in a heterostructure of $10 \times 3\sqrt{3}$ CrTe_2

stacking on $16 \times 4\sqrt{3}$ bilayer graphene (Fig. R5). We thus double checked our conclusions with the revised lattice and hetero-model (Figure R5) and found our previously believed magnetic ground state of the zigzag (ZZ) order is still valid with even more pronounced stability.

We specified the estimated in-plane compressive strain of 5% in a_1 and tensile strain of 3 % in the a_2 directions in the main text (page 8) and put those above discussions in Supplementary Figure S11 and Table S1

Table R1. Geometric and magnetic details of the free-standing and substrate-supported ML CrTe₂.

CrTe ₂ -1L	Mag. Config.	ΔE (meV/Cr)	a(Å)	Mag. Mom. (μ_B)	
				Cr	Te
Fully relaxed (U = 4.4 eV)	FM	55.3	3.7/3.7/3.7	3.29	-0.21
	sAFM-ABAB	29.0	3.6/3.8/3.8	3.28	-0.05
	sAFM-AABB	0.0	3.6/3.6/3.8	3.40	-0.21
	ZZ	17.6	3.6/3.6/3.7	3.34	-0.05
Free standing (U = 3.0 eV)	FM	25.5	3.7/3.7/3.7	3.11	-0.18
	sAFM-ABAB	6.0	3.5/3.7/3.7	3.03	-0.04
	sAFM-AABB	4.4	3.5/3.6/3.7	3.16	-0.06
	ZZ	0.0	3.6/3.6/3.7	3.08	-0.04
Hetero. Lattice (U = 3.0 eV)	FM	76.8	3.4/3.7/3.7	3.13	-0.15
	sAFM-ABAB	83.5		3.13	-0.04
	sAFM-AABB	59.9		3.18	-0.14
	ZZ	0.0		3.08	-0.04
On bilayer graphene substrate (U = 3.0 eV)	FM	71.0	3.4/3.7/3.7	3.15	-0.18
	ZZ	0.0		3.07	-0.06

Figure R5. Top and side views of the $10 \times 3\sqrt{3}$ ML CrTe₂ on the $16 \times 4\sqrt{3}$ bilayer graphene heterostructure model. The grey lines correspond to the graphene substrate in the top view for clarity.

Q15 The way how Fig. S11 is plotted suggests that the energy for the sAFM-AABB does not depend on the doping. Is this correct?

We appreciate the reviewer's concern and regret that our writing may not be clear enough. The original Fig. S11 (Figure R6a) plots relative total energies of three magnetic orders, i.e. FM, zigzag, sAFM-ABAB and sAFM-AABB, as a function of doping level, while the energy of the sAFM-AABB order was used as the reference zero at each doping level. It was intended to show that the sAFM-AABB order remains the ground state under charge doping in our previous calculations using $U = 4.4$ eV. In this revision, we replotted Fig. S11 (updated Figure S12) additionally with results revealed using $U=3.0$ eV and chose the zigzag order as the reference zero (Figure R6b). By using a more reasonable U value, our calculations indicate the zigzag order has the lowest total energy in a wide range of doping levels and should be robust in our experiments.

We replotted the figure and revised the caption of Fig. S12 clearly explaining that they are relative energies that more straightforwardly show the robustness of the zigzag order.

Figure R6. Relative total energies of the zigzag (green reverse triangles), sAFM-AABB (blue triangles) and sAFM-ABAB (red circles) orders as a function of the electron/hole doping level using $U = 4.4$ eV (a) and 3.0 eV (b) calculated with optB86b-vdW+UJ. The zigzag order was chosen as the reference zero.

Reviewer #2 (Remarks to the Author):

This work reports the study of monolayer CrTe₂. Main claims include: (1) growth of monolayer CrTe₂; (2) CrTe₂ has zigzag antiferromagnetic order with easy axis along the Cr-Te bonding direction; (3) observation of spin-flop type transition in real space. Bulk crystal CrTe₂ has been demonstrated as ferromagnet with high Curie temperature. The

reported results can be potentially very interesting if the claims are correct. Unfortunately, the current status of the paper is a bit premature and I cannot recommend it for publication.

Thank the reviewer for the potentially interesting results of our study. We appreciate it very much for the valuable suggestions and insightful critics over our manuscript, which definitely help us to improve the presentation of our work more clearly. Below are our point-to-point responses to the reviewer's issues.

(1) I am not convinced that there is even a magnetic order. Temperature dependence could provide useful information of magnetic phase transition. For example, how does the observed zigzag pattern depend on temperature? Ideally, the authors need to have a second approach to demonstrate the formation of magnetic order.

Thanks for raising this issue. We assign the zigzag pattern to a spin-dependent contrast based on following considerations. First, the zigzag pattern is not observable with a spin-averaged W tip. Second, the zigzag pattern is observable with a spin-polarized Cr tip and double-confirmed with an Fe tip. Third, the zigzag pattern responses to external magnetic field. Those experimental results compellingly indicate the spin-dependent contrast and directly evidence the existence of a static AFM magnetic order. For clarity, we have added a sentence "The spin contrast directly evidences the existence of an AFM magnetic order." on pages 7 of the main text.

We agree that it is ideal to measure the temperature dependence of the zigzag pattern in our SPSTM experiments. Those magnetic tips are, however, coated with Cr or Fe thin films, whose magnetization cannot be stabilized at high temperatures. It is thus extremely challenging to perform such measurements with the SPSTM approach. Nevertheless, we carried out another two approaches, i.e. SQUID (superconducting quantum interference diffraction) and SHG (second harmonic generation), to support the existence of the AFM order, the details of which could be found in the reply to Q3 of Referee 3. The SHG method suggests the Neel temperature in ML CrTe₂ up to 320 K.

To clarify these points, we have added relevant discussions in Page 13 and 14, and a new supplementary Figure S17.

(2) The growth produces both monolayer and bilayer. What do you see in bilayer? Is the zigzag pattern special to monolayer?

Thanks for raising this important issue. The zigzag spin contrast is only observed in monolayer. No spin contrast is observed in the second layer. The absence of spin contrast in the second layer is currently not well understood. However, we notice from the STM image of Fig. 1h that the 2×1 structure shows mixed orientations in the second layer, which is distinct to the monolayer. The measured lattice constants of the second layer CrTe₂ also change to $a_1=3.64 \text{ \AA}$, $a_2=3.61 \text{ \AA}$, $a_3=3.60 \text{ \AA}$. This contrasts with the lattice constants of the first layer, which are $a_1=3.4 \text{ \AA}$, $a_2=3.7 \text{ \AA}$, $a_3=3.7 \text{ \AA}$.

For clarity, we have added a sentence "It is noted that no spin contrast is observed in the second layer CrTe₂ at present, whose origin should be investigated in future." to Page

8.

(3) What is the magnetic response of Cr tip? In tunneling measurements, the spin structure of Cr will affect the characterization of CrTe₂. How do you characterize the Cr tip magnetization orientation? How do you experimentally conclude the spin orientation in CrTe₂?

Thanks for raising this issue. We agree that the magnetization of the Cr tip affects the characterization of CrTe₂. We characterized the Cr tip by imaging a standard magnetic sample of Co nano-islands on Cu(111) (Ref. PRL 92, 057202 (2004)), whose magnetization is out-of-plane. This can, however, only prove the tip has out-of-plane spin sensitivity, without information about whether the tip also has in-plane spin sensitivity. It is also worth to mention that the tip magnetization has finite possibility of change during tip approach to the sample. The observed zigzag spin pattern indicates both CrTe₂ and Cr tip have spin sensitivity.

The magnetization orientations of the Cr tip and CrTe₂ are determined in the following way. First, from Fig. Fig. S8, the spin contrast varies at different domains of CrTe₂ at zero magnetic field. This suggests both the tip and sample have in-plane component of magnetization. Second, the zigzag spin contrast reverses its phase under a small external field of 0.2 T (Fig. S6), that is applied perpendicular to the basal plane of CrTe₂. This demonstrates both the tip and sample should have out-of-plane component of magnetization at 0.2 T. The Cr tip magnetization hardly changes under small magnetic fields, whose switching threshold field is typically an order of magnitude larger, due to the antiferromagnetism of the Cr tip with strong magnetic anisotropy. Therefore, the observed spin contrast reversal comes from the sample, rather than the tip. This observation has been repeated with multiple Cr tips. Third, the magnetization of the CrTe₂ is not purely in-plane at zero magnetic field. Otherwise, one would expect the same phase of spin contrast between the ± 0.2 T. Thus, the magnetization orientation proposed in Fig. 3 is the most plausible scenario.

The magnetization orientation of CrTe₂ is also substantiated from the measurement with Fe-coated tips. The Fe-coated tip has in-plane magnetization at zero magnetic field, because of the surface and interface anisotropy of the coated ferromagnetic Fe film. The spin contrast of different domains also varies in Fig. S9a, confirming the in-plane spin component of CrTe₂. The Fe tip magnetization can be aligned to out-of-plane with a magnetic field applied perpendicular to the basal plane. Under a 3 T out-of-plane field, the spin contrast at the upper-right domain of Fig. S9b has similar zig-zag spin contrast as that of the lower-left domain. This conforms to the out-of-plane spin component of CrTe₂.

For clarity, we have added those discussions to a new Supplementary Note.

(4) What is the saturation field along the z direction? If the magnetic field (e.g. 5T) is larger than the saturation field, then all spins should be aligned in the Z direction. Then I do not expect to see a zigzag pattern under large magnetic field, as those in Fig. S5. In addition, at B field of 5T, the spin orientation of Cr tip will change as well. But the zigzag patterns in Fig. S5 does not change at all. This is confusing.

As we answered to Q3 of Reviewer 1 and Q3 of Reviewer 3, the critical field was theoretically estimated over 48 T using the spin-exchange energy of 76.8 meV/Cr and the MAE of 1.9 meV/Cr in the classical spin-flop picture, while the saturation field is even larger. This is consistent with our experimental data shown in Fig. S5 (now Fig. S6) that the zigzag spin pattern maintains under a field of 5 T.

In terms of the Cr tip, its spin orientation keeps unchanged up to 5 T in our experiment. It was reported that, the spin orientation of Cr tips, if well prepared, can sustain to high magnetic fields, e.g. 8 T in Nature 467, 1084–1087 (2010).

We mentioned the critical field over 48 T on page 10 of the revised main text and mentioned the tip issue in the caption of Fig. S6. We also added a phrase “and the magnetization direction of the tip sustains up to 5 T” in Page 7 of the main text.

(5) The authors observed domain effects (Fig. S7). Could the structural domain give rise to the observed domain pattern?

Thanks for raising this interesting issue. Domain effects are observed. Indeed, as is shown in Fig. S7 (now Fig. S8), the structural domain coincides with the magnetic domain.

For clarity, we have added a sentence “The structural domain coincides with the magnetic domain.” to the caption of Fig. S8.

Reviewer #3 (Remarks to the Author):

The authors present a study on CrTe₂ monolayer in which they use spin polarized scanning microscopy to conclude that the material is an antiferromagnet, and to probe a spin-flop transition that occurs in the presence of an applied magnetic field.

I have a number of substantial concerns about the work. Most of these concerns stem from what seems to be a lack of sufficient control experiments, or other types of sample interrogation which could corroborate the antiferromagnet order suggested by the authors.

We thank the reviewer for his/her valuable comments. We agree with the reviewer that other experiments should be conducted to corroborate the antiferromagnetic order and the spin-flop transition. Below are our point-to-point responses to the reviewers' concerns.

1. There should be control experiments provided on the tip which demonstrate that the tip magnetization is unaffected by the magnetic field. This seems to be the most obvious potential artifact, and it is simply asserted in the text that the tip is impervious to magnetic fields up to 2 T (without reference to previous work or to experimental data) (line 145). Also, the additional experiment using an Fe tip is used as a control. The text seems to imply that the stability of the Fe tip is more established, although I'm not sure. However, again no reference is provided. This seems quite unexpected to me, and it is crucial to conclusively establish that the tip is not introducing magnetic field-dependent artifacts. This concern is amplified given some of the aspects of the data, as described next.

Thanks for raising this important issue. The antiferromagnetic Cr tip has the advantage of its magnetization being resistant to external magnetic field conventionally up to 2 T. Such magnetization property of Cr tips has been discussed in Page 1504 of the reference Rev. Mod. Phys. 81, 1495 (2009) (Ref. 17 of the main text). For clarity, Ref. 17 is cited here again. We can also clearly evaluate whether the tip magnetization was preserved during the measurement. For Cr tips, its magnetization can be switched by magnetic fields. However, if that happens, the observed zigzag spin pattern would reverse its contrast abruptly with increasing magnetic field, which was not observed in Fig. S6. Moreover, the zigzag spin pattern recovers to its original contrast once the magnetic field is removed. All this information rigorously demonstrates the tip magnetization was preserved.

For Fe tips, its magnetization is in-plane at zero magnetic field, because of the surface and interface anisotropy of the coated ferromagnetic Fe films. Its magnetization can be aligned by external magnetic field, which provides complementary information to the Cr tips. Such magnetization character of Fe-coated tips has also been depicted in Page 1504 of the reference Rev. Mod. Phys. 81, 1495 (2009) (Ref. 17 of the main text). The corresponding author of our manuscript also has the same experience about the Fe tips (Ref. PRL 108, 087203 (2012), Nat. Commun. 3, 953 (2012)). For clarity, we have also added a reference as Ref. 17. The spin contrast of Fig. S9a recovers once the magnetic field is removed, demonstrating the magnetization of the Fe tip recovers to its original direction. It's worth to mention that this type of A-B-A experiments are always performed in spin-polarized STM measurements, to check whether the tip magnetization is changed or not.

For clarity, we have added discussions on the properties of the Cr and Fe tips in Supplementary Note.

2. The magnetic field-dependence does not seem consistent with a spin-flop transition. A spin-flop transition typically occurs at a nonzero threshold field, given by $(2 * H_{\text{exchange}} * H_{\text{anisotropy}})^{1/2}$. Using the density functional theory estimates of the anisotropy energy (on the order of 1 meV/Cr atom), and exchange energy (I come up with 60 meV/Cr atom as the energy cost of flipping one Cr atom direction in the staggered configuration), I estimate the spin-flop field is 55 T! This is a huge field, although the way I estimate exchange energy based on your DFT result is quite crude. (Also, as an aside, the DFT calculation of magnetocrystalline anisotropy can be very delicate, requiring very fine k-mesh. The authors should check that their results are well converged - I estimate the anisotropy field computed as 5 T, which is quite large.) Regardless of the value inferred from DFT, the experimental data seems to show that the signal ascribed to spin-flop evolves smoothly from zero field. In other words, there's no threshold field, and there should be. There should be. Returning to numbers: for CrI₃, $H_{\text{anisotropy}} = 1.7$ T and $H_{\text{exchange}} = 0.76$ T, resulting in spin-flop field of 1.6 T (<https://arxiv.org/abs/2001.04044>).

We highly appreciate the reviewer's comment and suggestions which do help with improving our understanding of this material.

We first examined the convergence of the adopting k-mesh. Table R2 shows the trend of k-convergence of MAE. While the previously used 10x6x1 mesh nearly reaches the convergence, the density of it needs to slightly increase to, at least 14x8x1, to reveal a converged energy difference of ~0.1 meV. We thank the reviewer for raising this issue and double-checked other energy values using the denser mesh.

Table R2. K-mesh convergence of MAE of free-standing ML CrTe₂ based on updated lattice constant using U = 3.0 eV.

CrTe ₂ -1L-Zigzag	Mag. Axis	MAE (meV/Cr)			
		10×6×1	14×8×1	18×10×1	20×12×1
Free standing	x	0.83	0.92	0.87	0.90
	y	0.16	0.13	0.12	0.12
	z	1.96	1.91		
	O _o	0.00	0.00	0.00	0.00
	O _{i1}	0.80	0.82		
	O _{i2}	1.53	1.45		

We next checked the effects of the updated lattice constant and slightly reduced U on the magnetic ground-state and anisotropy. The reduced U does not change the conclusions qualitatively and the updated lattice constant keeps the original easy axis. More candidate magnetization directions in the y-z plane were considered in the revision (Fig. R7a and R7b). The easy axis of the zigzag order was mis-labeled as pointing in the Cr-Te bonding direction (red dashed arrow) in our previous manuscript, which was revised to the direction indicated with the blue dashed arrow (O_o). This direction is 70° off the z axis and it only needs ~0.1 eV to rotate it by 20°.

Given an exchange energy of 76.8 meV/Cr of ML CrTe₂ and the converged MAE of 1.91 meV/Cr (energy difference between magnetization along O_o and z direction), we derived a spin-flop field of 48 T based on the classical spin-flop picture under vertical magnetic field. Including substrate in our calculations would slightly increase the spin-flop field to 49.1 T ($H_E = 71.0$ meV and $H_A = 2.16$ meV/Cr, Supplementary Table S1 and S2). This field is over an order of magnitude larger than the observed field of 0.2~0.5 T and thus suggesting another flop picture, perhaps a non-collinear one, rather than the classical one.

Figure R7. Magnetic anisotropic energy (MAE) of ML CrTe₂. (a) Magnetization axes considered in the MAE calculation. x , y and z axes correspond to the directions of the lattice vectors. A Cr-Te plane is marked with a partially transparent orange rectangular. (b) Angular dependence of the MAE of ML CrTe₂ with the direction of magnetization lying on yz plane. The energy of magnetic moments align along O_o is set to zero as reference.

Figure R8. Illustration of the rotation of magnetic moments under out-of-plane field. (a) Magnetization axes considered in the MAE calculation. Here blue dashed arrow corresponds to the easy axis O_o . (b) The green and red arrows indicate the magnetic moments M_1 and M_2 of two anti-ferromagnetically coupled adjacent Cr atoms. The dashed arrows correspond to the decomposed magnetic moments component along the z and y axes. (c-d) Evolution of the z -component (c) and y -component (d) of magnetic moments under a vertical field.

As discussed above, the O_o direction with 70 degrees off the z axis is the easy axis direction of ML CrTe₂ (Figure R8a), leading to a different spin-flop picture (Ref. npj Computational Materials 2, 16032(2016)). As shown in Figure R8b (updated Fig. 3e), we can decompose the magnetic moment of each Cr atom into two components parallel to the y and z axes, denote M_z and M_y , respectively, which do rotate or flop as a function of the magnetic field along z . We then discuss the field-dependent moment rotation using two adjacent and anti-parallel oriented moments, namely M_1 and M_2 .

In terms of M_z (along z , Fig. R8c, updated Fig. 3f), a growing magnetic field parallel to the z axis causes moments M_{1z} and M_{2z} flopping to the y direction, which is the second

easy axis with a MAE of 0.12 meV/Cr, corresponding to a threshold field of~ 0.35 T and consistent with the small experimental critical field of 0.2 T. The spin-flop transition of z-component magnetization would also lead to a redistribution of charge density on Te atoms, leading to the SPSTM observed spin-contrast reversal.

In the M_y case, moments M_{1y} and M_{2y} gradually rotate to the z direction with a growing magnetic field along z, thus leading to a non-collinear magnetic configure that shows an emerging net magnetic moment along the z direction (Fig. R8d, updated Fig. 3g). Our non-collinear constrained DFT calculation reveals that a 2-degree rotation of magnetic moment towards z only requires a moderate energy cost of 0.15 meV/Cr, but the energy cost speedy increases for larger rotation angles (Supplementary Figure S13a and Fig. R9a).

We thus estimated the magnetic moment evolution under an increasing magnetic field as shown in Figure R9b (S13b). Because of the emergence of the non-collinear order, the net magnetization along z direction dramatically increases under a small magnetic field of no more than 0.8 T. The increasing rate is reduced under higher magnetic fields, which is consistent with our results of out-of-plane m - H curves taken with SQUID (Figure R10a). Domain boundaries in our samples could also give rise to the higher increasing rate under small magnetic fields. However, complete rotation of the moment to the z direction needs a much larger energy to surmount the intralayer spin-exchange energy of 76.8 meV/Cr; this requires a magnetic field over 48 T, consistent with our observed unsaturated magnetization under a few Tesla.

To clarify these points, we revised relevant sentences on pages 10 and 11 and Figure 3 of the main text, as well as Figs. S13 and Tables S2 and S3 for clarity.

Figure R9. (a) Angular dependent relative energies of non-collinear orders. The energy the magnetic ground state ZZ is set to zero as reference. (b) Theoretical estimated net magnetic moments per supercell under magnetic field.

3. Given all of the above, it would be quite worthwhile to perform additional experiments on this material to find other evidence of spin-flop transition. For example, you can do XMCD, or squid measurements to detect the formation of magnetization associated with this state (as in <https://journals.aps.org/prb/pdf/10.1103/PhysRevB.101.094429>). Also, measurements to estimate the Neel temperature would be useful, depending on how

conclusive things look after additional control measurements described above.

Thanks for raising this important issue. Following the reviewer's suggestion, we have performed SQUID measurements to our sample. As is shown in Fig. R10(a), the magnetic moment (m) of ML CrTe₂ increases monotonically from zero value with increasing out-of-plane magnetic field (H). Above about 0.3 T, the increasing rate of magnetization becomes significantly reduced, which agrees with our spin-resolved spectra in Fig. 4(f). With increasing temperature, the critical field for the slope change in the m - H curves becomes smaller. Concomitantly, the increasing rate of m after the critical field also becomes reduced. All these observations demonstrate the existence of AFM order in our sample. From our SQUID measurement, we estimate the Neel temperature may be higher than 300 K. To evaluate the spin-flop transition, we zoom-in the m - H curve around the zero field. As is shown in Fig. R10(b), the slope of the m - H curve obviously changes at about 0.3 T, which is associated with the spin-flop transition field. This conforms to our SPSTM measurement. It is noted that our magnetic field can be only applied out-of-plane, which has an angle of 70° relative to the easy axis. Therefore, we could only observe a change in slope of m - H curve around the spin-flop field, which is similar to Ref. PRB 76, 134415 (2007).

Moreover, we have further measured our sample with second harmonic generation (SHG) method. Since the ML CrTe₂ has inversion symmetry in its crystallographic structure, the SHG signal would come from the time-reversal symmetry breaking, i.e. the AFM order. This technique has been used to determine the AFM order in van der Waals materials by some of our coauthors in Ref. Nature 572, 497-501 (2019). As is seen in Fig. R10(c), the SHG signal decreases with increasing temperature, until a tendency of saturation at a finite value above 320 K. This contrasts with the substrate, whose SHG signal is negligibly small, and invariant with temperature. The finite SHG signal of ML CrTe₂ above 320 K could possibly from its AFM signal or from the inversion symmetry breaking at the CrTe₂/graphene interface. This demonstrates the AFM order is no less than 320 K, consistent with the high Neel temperature estimated from the SQUID measurement.

For clarity, we have added a new supplementary Fig. S17 and the relevant discussions on pages 13 and 14 of the main text.

Fig. R10 SQUID and SHG measurements on ML CrTe₂. (a) Out-of-plane m - H curves of ML CrTe₂ taken with SQUID at different temperatures, showing AFM order. (b) Out-of-plane m - H curves of ML CrTe₂ taken with SQUID at 5 K, showing spin-flop transition. (c) Temperature dependent SHG measurements of ML CrTe₂ and the substrate.

Reviewers' Comments:

Reviewer #1:

Remarks to the Author:

Review of NCOMMS-20-46354A (revised manuscript)

Spin mapping of intralayer antiferromagnetism and spin-flop transition in monolayer CrTe₂ by J.-J. Xian et al.

In the revised manuscript, the authors have answered and clarified all my concerns and doubts. They provided various additional (experimental and theoretical) data and information that significantly improved the manuscript. I recommend publication in Nature Communications.

Reviewer #2:

Remarks to the Author:

I have read through authors' response to the comments. The revised manuscript is indeed improved. However, I am not convinced by the claim of the zigzag AFM order and the interpretation. In fact, the observed spin contrast is far from direct evidence of any magnetic order.

(1) Zigzag AFM order should be decoupled from the magnetic field, unless the field is large enough to flip all spin polarization. It does not make sense that adjacent zigzag spin chain flips direction by flipping the magnetic field. In this case, the author claims a net moment developed in the Z direction. In order to explain the data, the authors have to introduce a spin rotation of My to Z direction which has a nonlinear dependence on the rotation angle. I am not sure how realistic this process is since I do not have way to check. In addition, line 227, the authors claim the estimated Mz flips to Y direction at about $\sim 0.35T$, which is consistent with the experimental observed critical field of 0.2T. However, the flip of Mz to Y does not introduce a net contrast in the Z direction. So this part is even correct, it cannot explain the data.

(2) Could the authors plot the spin contrast vs magnetic field? If the nonlinear spin-rotation of My to Z axis is correct, the spin contrast should show this process.

(3) It is critical to know what happens below 0.2T. The authors need to show the data at even smaller magnetic fields. Ideally, the authors need to show the spin contrast vs magnetic field from -0.2 to +0.2, as the magnetic field sweeps back and forth. This set of data would help to reveal the origin of the observation.

(4) It seems spin contrast is already there at 0T (Fig. 2a). This is opposite to the expectation from an AFM state.

(6) I do not understand why the bright zigzag chain shifts as the magnetic field flips sign. If the physical picture in Fig.3 is correct, there should be no spatial shift of the spin-chain in the lateral direction.

Reviewer #3:

Remarks to the Author:

Overall I think the authors have provided a fairly robust response to my concerns. I have a couple of remaining comments / questions for the authors to consider. After some further refinement and clarifications, I believe the paper may now be published.

1. Given that the charge density changes with magnetic order (as shown in Fig. 3(h,i,j), Fig. 4), shouldn't a non-magnetic tip be sensitive to magnetic order (see line 138, which states that the order is insensitive to W tip)?

2. The authors should reconsider use of "spin-flop" terminology. The behavior observed in the experiment is a continuous canting of the AFM spins. Typically the term "spin-flop" is reserved for threshold-type behavior when the applied field is aligned to an easy axis (i.e., aligned to the antiferromagnet Neel vector), and the Neel vector abruptly transitions to a perpendicular-to-field orientation. In this case, the field is closer to being perpendicular to the easy axis/Neel vector, and the moments of the AFM spin continuously cant along the B field as its magnitude increases. I

would then just describe this as "canting" of the AFM spins.

3. The behavior of $m(T)$ may largely be understood in terms of the complex energy landscape of the magnetic anisotropy. For example, consider the statement around line 296: "the net magnetization along z dramatically increases under a small magnetic field of no more than 0.8 T because of the emergence of the non-collinear order and the increasing rate is also reduced at higher magnetic fields." This could be framed in a more explicit way if one can recognize the nonlinear response of m as a consequence of the "soft" potential near O^o , which becomes increasingly hard as one deviates further from the minimum. It's hard to determine if this is exactly correct, given the data provided, but the authors may be able to do more analysis see if this viewpoint is instructive.

4. I don't understand statement that y -axis is an easy axis: the data of Fig. 3(d) does not seem to show that the y -axis is a local minimum of energy. Also, given point 3, it may be worthwhile to compute more points in the energy versus Neel vector orientation curve (inset of Fig. 3d).

5. The shape of the computed $m(B)$ curve Fig. S13(b) looks qualitatively similar to the experimental SQUID data Fig. S17. It seems as though quantitative comparison is possible - how do the numbers compare?

Reply to Review

Reviewer #1 (Remarks to the Author):

Review of NCOMMS-20-46354A (revised manuscript)

Spin mapping of intralayer antiferromagnetism and spin–flop transition in monolayer CrTe₂

by J.-J. Xian et al.

In the revised manuscript, the authors have answered and clarified all my concerns and doubts. They provided various additional (experimental and theoretical) data and information that significantly improved the manuscript. I recommend publication in Nature Communications.

Thank the reviewer for recommending the publication of our manuscript.

Reviewer #2 (Remarks to the Author):

I have read through authors' response to the comments. The revised manuscript is indeed improved. However, I am not convinced by the claim of the zigzag AFM order and the interpretation. In fact, the observed spin contrast is far from direct evidence of any magnetic order.

Thank the reviewer for recognizing the improvement of our manuscript. The reviewer's remaining major concerns are arisen from two fundamental concepts regarding the spin-flop transition and the principle of SPSTM, which are actually well-established. Below are our point-to-point responses to the reviewer's concerns. It could be helpful to read from our reply to Questions 1 & 4 first.

(1) Zigzag AFM order should be decoupled from the magnetic field, unless the field is large enough to flip all spin polarization. It does not make sense that adjacent zigzag spin chain flips direction by flipping the magnetic field. In this case, the author claims a net moment developed in the Z direction. In order to explain the data, the authors have to introduce a spin rotation of M_y to Z direction which has a nonlinear dependence on the rotation angle. I am not sure how realistic this process is since I do not have way to check. In addition, line 227, the authors claim the estimated M_z flips to Y direction at about $\sim 0.35T$, which is consistent with the experimental observed critical field of $0.2T$. However, the flip of M_z to Y does not introduce a net contrast in the Z direction. So this part is even correct, it cannot explain the data.

If we understood correctly, the reviewer's concerns could be fully addressed by explaining the difference between the spin-flop and spin-flip transitions.

For the spin-flip transition, the AFM sublattice magnetization is invariant against an applied magnetic field parallel to the sublattice magnetization (\$H_{\parallel}\$ ), until a critical value, \$H_c\$, after which all the magnetization flips to the magnetic field direction (Fig. R1a). This case

is exactly what the reviewer described, which occurs for AFM magnets with strong magnetic anisotropy.

On the other hand, for AFM magnets with weaker magnetic anisotropy, as in our case, a spin-flop transition occurs with two critical fields, namely H_{SFO} and H_{C} , involved in the presence of H_{\parallel} . At H_{SFO} , the invariant AFM sublattice magnetization under smaller fields becomes perpendicular to H_{\parallel} , which further cants towards H_{\parallel} under fields beyond H_{SFO} and reaches a saturation at H_{C} (Fig. R1b).

Nevertheless, for a magnetic field applied perpendicular to the magnetization direction (H_{\perp}), AFM magnets showing either the spin-flip or the spin-flop transition have their sublattice magnetizations tilted towards H_{\perp} right after the field is applied (Fig. R1c). [Ref. Mathias Getzlaff, Fundamentals of Magnetism, Springer (2008), Pages 100-102.]

Our calculation shows the easy axis of magnetic anisotropy of CrTe_2 is along the O_o direction, which is 70° off the z axis. Since our magnetic field is applied along z , we decomposed the total magnetization into the y and z directions, i.e. M_y , and M_z . Under an out-of-plane magnetic field, M_z flips to y at H_{SFO} and meanwhile M_y cants towards z , leading to a net magnetization in the z direction. If the field is over H_{SFO} , both M_z and M_y continuously rotate towards z , until being saturated at H_{C} . The DFT-revealed relationship of magnetization and magnetic field during such a process agrees with the SQUID measurement. And, such a spin-flop scenario is also consistent with the spin contrast observed by SPSTM, whose details are formulated in our reply to Question 6.

SPSTM can resolve an AFM spin contrast without the need of generating net magnetization along z . Please see our reply to Question 4 for more detailed explanations. As depicted on Page 6 of our main text, our tip magnetization is canted with respect to the z direction, endowing it with in-plan spin sensitivity. Thus, even if M_z flops to the y direction, it can still be resolved in SPSTM.

Fig. R1. Schematics of spin-flip and spin-flop transitions. The lower panels are their magnetization (M) as a function of the magnitude of the magnetic field (H). The upper panels depict their spin configurations at corresponding H values. When the magnetic field is applied parallel to the AFM sublattice spins, M_+ and M_- , their magnetization reaches saturation (M_S) via a spin-flip process (a) [spin-flop process (b)] for AFM magnets with

strong (weak) magnetic anisotropy. When the magnetic field is applied perpendicular to the AFM sublattice spins, their magnetizations indicate spin canting towards the field direction for AFM magnets undergoing either the spin-flip or spin-flop transitions (c).

(2) Could the authors plot the spin contrast vs magnetic field? If the nonlinear spin-rotation of My to Z axis is correct, the spin contrast should show this process.

To quantitatively evaluate the spin contrast, we performed FFT analysis of the SPSTM images under different magnetic fields. As shown in Fig. R2a, the spin contrast reverses with the magnetic field direction switching, as is indicated with a defect marker labelled in a green circle, which is in line with the data set shown in Fig. 2 of the main text. Moreover, the spin contrast intensity becomes more enhanced with increasing magnetic field strength in both directions. The spin contrast intensity can be quantitatively evaluated from the ratios between FFT patterns originating from the spin contrast and the atomic resolution of the crystal lattice, as has been applied in Ref. Science 345, 653 (2014). Namely, a more enhanced spin contrast produces a larger ratio in its corresponding FFT image. As an example, Fig. R2b displays the FFT image extracted from the SPSTM image at 1 T in Fig. R2a, where the diffraction spots of the spin contrast (q_{AFM}) and the crystal lattice (q_{Te}) are indicated. We plot the spin contrast intensity, depicted by such a ratio vs magnetic field, as shown in Fig. R2c. The spin contrast intensity increases with increasing magnetic field, with a minimum at about -0.2 T, the reversal field of spin contrast, whose details are depicted in our reply to Question 3. Please also refer to our reply to Question 6 for further discussion of the figure.

We note that the spin contrast intensity depends on the relative alignment of magnetizations between the tip and sample, as well as their spin polarizations at corresponding energies. Since the tip magnetization and spin polarization cannot be determined precisely, the spin contrast intensity cannot deliver the absolute magnetization of the sample. Nevertheless, the spin contrast intensity plotted in Fig. R2c already shows dependence on the magnetic field strength, conforming to the rotation of sample magnetizations within a spin-flop transition picture.

Fig. R2. Spin contrast variation of monolayer CrTe₂ under large magnetic fields. (a) SPSTM images ($V_b = 30$ mV, $I_t = 200$ pA) of monolayer CrTe₂ under different magnetic fields. A defect marker is marked with a green circle in each image. (b) FFT image of SPSTM on monolayer CrTe₂ at 1 T. The diffraction spots from the Te lattice (q_{Te}) and the AFM spin contrast (q_{AFM}) are shown. (c) FFT intensity of q_{AFM} relative to q_{Te} under different magnetic fields. The data for positive (negative) fields is presented in black (red). The blue arrows mark the magnetic field sweep history.

(3) It is critical to know what happens below 0.2T. The authors need to show the data at even smaller magnetic fields. Ideally, the authors need to show the spin contrast vs magnetic field from -0.2 to +0.2, as the magnetic field sweeps back and forth. This set of data would help to reveal the origin of the observation.

By following the reviewer's request, we plot the spin contrast intensity vs magnetic field at smaller magnetic fields with different sweep directions, as shown in Fig. R3. The spin contrast intensity is acquired in a same strategy as depicted in the reply to Question 2. We only show the SPSTM images between -0.3 T and +0.3 T to evaluate the spin contrast reversal. As shown in Fig. R3a, the defect marker labeled in the green circle is on a bright zigzag chain for $H > -0.2$ T, which becomes in between the zigzag chain for $H < -0.2$ T. This demonstrates that a spin contrast reversal occurs at -0.2 T. The relative FFT intensities shown in Fig. R3b indicates the spin contrast intensity appears the lowest around the spin contrast reversal field of -0.2 T.

Fig. R3. Spin contrast variation of monolayer CrTe₂ under small magnetic fields. (a) SPSTM images ($V_b = 30$ mV, $I_t = 200$ pA) of the same location of monolayer CrTe₂ at different magnetic fields. A defect marker is marked with a green circle in each image. (b) FFT intensity of q_{AFM} relative to q_{Te} under different magnetic fields. The FFT intensities are extracted from the SPSTM images shown in (a). The data for positive (negative) fields is presented in black (red). The history of magnetic field application is depicted with black (blue) arrows in (a) [(b)]. This data set was taken at the same area as that of Fig. R2.

(4) It seems spin contrast is already there at 0T (Fig. 2a). This is opposite to the expectation from an AFM state.

The reviewer's concern appears relevant with the principle of SPSTM. SPSTM utilizes spin-dependent electron tunneling effect to achieve spin contrast. For SPSTM measurements, the spin-polarized tunneling current (I_{SP}) is sensitive to the relative angle (ϕ) between the magnetizations of the tip (P_{tip}) and the sample (P_{sample}), describing as $I_{SP} \propto P_{tip} \cdot P_{sample} \cdot \cos\phi$ [Please see Page 1500 of Ref. Rev. Mod. Phys. 81, 1495 (2009)]. In

particular, the tip-sample magnetizations with the parallel configuration offer a larger tunneling current than their antiparallel counterpart. Therefore, the variation of tunneling current caused by the different magnetization alignments gives rise to the spin contrast in SPSTM images (Fig. R4a).

For AFM magnets, they do not exhibit global magnetization, as the reviewer mentioned. However, they have local magnetic moments at the atomic scale with opposite sublattice magnetizations at 0 T. SPSTM can deliver spin resolution for AFM magnets due to its capability of achieving spin-dependent tunneling with atomic resolution (Fig. R4b). This capability of resolving AFM order has been demonstrated in previous studies in AFM metals [Science 288, 1805 (2000)], Fe-chalcogenides [Science 345, 653 (2014), Nat. Commun. 8, 14074 (2017)] and Ir-oxides [Nat. Phys. 15, 1267 (2019)].

Thus, an AFM magnet / state could show the spin contrast of Fig. 2a at 0 T.

Fig. R4. Schematics of SPSTM and its resolving AFM spins. (a) Principle of SP-STM: the spin-polarized tunneling current flowing between a magnetic tip and a magnetic sample depends on the relative alignment of the local magnetization of tip and sample as well as on the spin polarization of the electronic states of tip and sample contributing to the tunneling current. [Adapted from Fig. 5 of Ref. Rev. Mod. Phys. 81, 1495 (2009)]. (b) Schematics showing resolving AFM sublattice spins (red and green arrows) with SPSTM. The thick (thin) black arrows depict large (small) tunneling current.

(5) We do not see question 5 in the report being passed from the editors.

(6) I do not understand why the bright zigzag chain shifts as the magnetic field flips sign. If the physical picture in Fig.3 is correct, there should be no spatial shift of the spin-chain in the lateral direction.

The schematics shown in Fig. R5 depict the tip magnetization and the spin densities of the CrTe₂ monolayer under magnetic fields. The tip magnetization is kept unchanged during the variation of magnetic fields, because of the AFM ground-state of tip with strong magnetic anisotropy. The AFM state in the CrTe₂ monolayer has a relatively weaker

magnetic anisotropy and thus undergoes a spin-flop transition under the applied magnetic field.

Fig. R5 (a) Top view of the spin density of ML CrTe₂ in zigzag order. (b-d) Schematics showing magnetization configurations of the tip and the monolayer CrTe₂ under different magnetic fields. The red and green arrows depict the magnetization directions of the Cr atoms and Te atoms.

In SPSTM measurements of the CrTe₂ monolayer, variation of spin-polarized tunneling current is dominated by the Te-*p* orbitals because the current exponentially decays with the distance between the wavefunctions of the tip and the sample. The two spin-components of each *p* orbital of a Te atom are anti-parallel polarized by the three nearest Cr atoms (marked with dashed circles in Figure R5a) and thus categorized into two groups, namely, $p_{x/y}$ (red bagel-like density on the Te atom) and p_z (green dumbbell-like density on the Te atom) orbitals, with opposite spin polarizations (indicated by red and green colors).

For simplicity, we assume the tip magnetization is parallel to the blue arrows in Figure R5 b-d. This assumption gives a small angle between the magnetization directions of the tip and those Te orbitals colored in green, and hence the spin-polarized tunneling current maximizes when probing the densities of those green-colored Te orbitals. The green-colored p_{xy} orbitals appear brighter in spin contrast at 0 T (Fig. R5b) because of its larger extension along *z* (marked with blue dashed line) than that of the green-colored p_z orbital (marked with red dashed line). Here, we used orange and light-blue transparent parallelograms to represent the Te rows showing brighter and darker spin contrasts, respectively.

Under a magnetic field applied along the +*z* direction (Fig. R5c), the magnetic moments of Cr, marked using green (red) arrows on Cr atoms, rotate anticlockwise (clockwise) to produce a net magnetization along the +*z* direction. As a consequence, their anti-parallel locked moments on Te concertedly rotate, showing a net out-of-plane magnetization but opposite in-plane moments. This makes the magnetization of the green-coloured p_{xy} (p_z) orbital nearly parallel (orthogonal) to the tip magnetization. The spin contrast consequently becomes more enhanced and maintains its phase relative to that of the 0 T.

When the field is applied along the -*z* direction (Fig. R5d), a reversed rotation occurs to the Cr magnetizations and their associated Te orbitals. The magnetization of the green-coloured Te- p_z (p_{xy}) orbital tends to be parallel (orthogonal) to the tip magnetization, causing spin contrast reversal compared to 0 T. Correspondingly, the spin contrast is minimal at the spin-reversal field, and becomes enhanced for fields away from the spin-reversal field, conforming to the observation in Figs. R2c and R3b.

To clarify these points, we revised relevant sentences about field dependent contrast reversal on page 12 and Fig. 3(h-j) of the main text and Supplementary S15 for clarity.

Reviewer #3 (Remarks to the Author):

Overall I think the authors have provided a fairly robust response to my concerns. I have a couple of remaining comments / questions for the authors to consider. After some further refinement and clarifications, I believe the paper may now be published.

1. Given that the charge density changes with magnetic order (as shown in Fig. 3(h,i,j), Fig. 4), shouldn't a non-magnetic tip be sensitive to magnetic order (see line 138, which states that the order is insensitive to W tip)?

Thanks for raising this issue. The three p orbitals of the Te atoms are categorized into p_{xy} and p_z ones with opposite spin polarizations, respectively. The magnetic field driven redistribution of charge density occurs between different spins and orbitals, making the spin-contrast reversal easier to be observed by a Cr-coated magnetic tips.

While using a non-magnetic W-tip, both the p_{xy} and p_z orbitals of a Te atom could be detected simultaneously, which does not show spin resolution and has blurry orbital-resolution. The eliminated or/and blurry resolutions thus lead to inappreciable changes of charge density between different spins or among different orbitals in STM images acquired using the W-tip.

To clarify these points, we added relevant discussions about the non-magnetic tips on page 12 of the main text for clarity.

2. The authors should reconsider use of "spin-flop" terminology. The behavior observed in the experiment is a continuous canting of the AFM spins. Typically the term "spin-flop" is reserved for threshold-type behavior when the applied field is aligned to an easy axis (i.e., aligned to the antiferromagnet Neel vector), and the Neel vector abruptly transitions to a perpendicular-to-field orientation. In this case, the field is closer to being perpendicular to the easy axis/Neel vector, and the moments of the AFM spin continuously cant along the B field as its magnitude increases. I would then just describe this as "canting" of the AFM spins.

Thanks for the suggestion. As shown in Figs. 3f and 3g, the evolution of the z-component and y-component of the magnetic moments under a vertical magnetic field can be seen as a typical "spin-flop" and "canting" transitions, respectively. Given the easy axis O_0 direction is in the yz plane and 70° off the z axis, the y-component of the magnetic moments is over 2.5 times larger than that of z-component. Under a magnetic field of over 0.2 T (the critical field of the spin-flop transition), the canting transition would dominate the magnetic evolution until the saturation is achieved.

Following the reviewer's suggestion, we thus revised the term "spin-flop" as "spin

reorientation” in this revision.

3. The behavior of $m(T)$ may largely be understood in terms of the complex energy landscape of the magnetic anisotropy. For example, consider the statement around line 296: "the net magnetization along z dramatically increases under a small magnetic field of no more than 0.8 T because of the emergence of the non-collinear order and the increasing rate is also reduced at higher magnetic fields." This could be framed in a more explicit way if one can recognize the nonlinear response of m as a consequence of the "soft" potential near O^o , which becomes increasingly hard as one deviates further from the minimum. It's hard to determine if this is exactly correct, given the data provided, but the authors may be able to do more analysis see if this viewpoint is instructive.

We highly appreciate the reviewer's comment and suggestions which do help with improving our understanding of this material. In accordance with the reviewer's suggestion, we carried out additional calculations to more closely examine the angular dependent relative energies of the non-collinear orders, which gives the relationships of rotation angle under magnetic field (Fig. R6a) and of magnetization along z with respect to rotation angle (Fig. R6b). Both plots derive a more detailed evolution relation between magnetic moment and magnetic field (Fig. R6c). A significant decrease of the slope, highlighted with the blue dotted lines in Fig. R6c, could be observed at smaller fields, which confirmed the "softer" potential near the anti-parallelled zigzag ground states as mentioned by the reviewer.

To clarify these points, we revised relevant sentences about the evolution of the magnetic moments on page 11 of the main text and added Supplementary Fig. S15 for clarity.

Fig. R6. The rotation of magnetic moments under out-of-plane field. (a) Theoretical estimated rotation degree of magnetizations moments direction under magnetic field from -3 T to 3 T. (b) The magnetization along z as a function of the rotation degree of magnetic moments. (c) Evolution of the magnetization along z under external magnetic field. Blue dashed lines represent the slope of the net magnetic moments.

4. I don't understand statement that y -axis is an easy axis: the data of Fig. 3(d) does not seem to show that the y -axis is a local minimum of energy. Also, given point 3, it may be worthwhile to compute more points in the energy versus Neel vector orientation curve (inset of Fig. 3d).

We thank the reviewer for raising this issue and regret that our discussion on the easy axis direction was not clear enough. Our calculations indicate the easy axis of magnetic anisotropy is along the O_o direction, which is in the yz plane and 70° off the z axis. We did not intend to claim the y -axis is the easy axis.

By following the reviewer's suggestion, we added more points in the MAE calculations and calculated the angular distribution of MAE, which could be described using spherical coordinates. Angles ϕ and θ ranging from 0° to 180° correspond to the angles between the magnetization direction and the z and x axes respectively (Figure R7a). As shown in Figure R7b, we confirmed that our conclusion on the easy axis direction of O_o (marked with white dashed circle) is robust.

To clarify these points, we revised relevant sentences about the MAE on page 10 and Fig. 3d of the main text and added Supplementary Fig. S14 for clarity.

Fig. R7. Angular dependence of the calculated MAE. Here, angles ϕ and θ correspond to the angles between the magnetization direction and the z and x axes, respectively. A step size of 10° is used in our calculations. The total energy of a configuration where the Cr moment is oriented to the O_o direction was chosen as the zero energy reference.

5. The shape of the computed $m(B)$ curve Fig. S13(b) looks qualitatively similar to the experimental SQUID data Fig. S17. It seems as though quantitative comparison is possible - how do the numbers compare?

We thank the reviewer for such a nice suggestion. For the experimental SQUID data, while the shape of the curves obtained in different measurements are similar, their exact values are not identical. It is presumably caused by the different distance between the sample and the SQUID probe installed in each measurement, which is particularly critical for monolayer samples with weak signals. This makes a quantitative comparison with the theory infeasible.

Reviewers' Comments:

Reviewer #2:

Remarks to the Author:

I have read through the comments and the authors made reasonable revisions. I am still not fully convinced by the conclusion because the interpretation heavily relies on the modelling.

Nevertheless, I recommend it for publication since the work will draw strong interest from the community.

Reviewer #3:

Remarks to the Author:

After reading the revised manuscript and the authors' responses to the referees, I believe the manuscript can be published in Nature Communications.

I have a small suggestion to the authors: in their presentation of the MAE data (Fig. 3d), they use an unconventional notation, in which ϕ and θ , represent polar and azimuthal angles, respectively. I would suggest they use conventional notation, where θ is polar and ϕ is azimuthal.

Reply to Review

REVIEWERS' COMMENTS

Reviewer #2 (Remarks to the Author):

I have read through the comments and the authors made reasonable revisions. I am still not fully convinced by the conclusion because the interpretation heavily relies on the modelling. Nevertheless, I recommend it for publication since the work will draw strong interest from the community.

Thank the reviewer for recommending the publication of our manuscript.

Reviewer #3 (Remarks to the Author):

After reading the revised manuscript and the authors' responses to the referees, I believe the the manuscript can be published in Nature Communications.

I have a small suggestion to the authors: in their presentation of the MAE data (Fig. 3d), they use an unconventional notation, in which ϕ and θ , represent polar and azimuthal angles, respectively. I would suggest theu use conventional notation, where θ is polar and ϕ is azimuthal.

Thank the reviewer for recommending the publication of our manuscript. We have revised the notation of angles, as suggested by the reviewer.